# SEMIRETRO: SEMI-TEMPLATE FRAMEWORK BOOSTS DEEP RETROSYNTHESIS PREDICTION

## ABSTRACT

Retrosynthesis brings scientific and societal benefits by inferring possible reaction routes toward novel molecules. Recently, template-based (TB) and template-free (TF) molecule graph learning methods have shown promising results to solve this problem. TB methods are more accurate using pre-encoded reaction templates, and TF methods are more scalable by decomposing retrosynthesis into subproblems, i.e., center identification and synthon completion. To combine both advantages of TB and TF, we suggest breaking a full-template into several semi-templates and embedding them into the two-step TF framework. Since many semi-templates are reduplicative, the template redundancy can be reduced while the essential chemical knowledge is still preserved to facilitate synthon completion. We call our method SemiRetro and introduce a directed relational graph attention (DRGAT) layer to extract expressive features for better center identification. Experimental results show that SemiRetro significantly outperforms both existing TB and TF methods. In scalability, SemiRetro covers 96.9% data using 150 semi-templates, while previous template-based GLN requires 11,647 templates to cover 93.3% data. In top-1 accuracy, SemiRetro exceeds template-free G2G 3.4% (class known) and 6.4% (class unknown). Besides, SemiReto has better interpretability and training efficiency than existing methods.

## 1 INTRODUCTION

Retrosynthesis prediction (Corey & Wipke, 1969; Corey, 1991) plays a crucial role in synthesis planning and drug discovery, which aims to infer possible reactants for synthesizing a target molecule. This problem is quite challenging due to the vast search space, multiple theoretically correct synthetic paths, and incomplete understanding of the reaction mechanism, thus requiring considerable expertise and experience. Fortunately, with the rapid accumulation of chemical data, machine learning is promising to solve this problem (Coley et al., 2018; Segler et al., 2018). In this paper, we focus on the single-step version [1]: predicting the reactants from a given product.

Common deep-learning-based retrosynthesis works can be divided into template-based (TB) (Coley et al., 2017b; Segler & Waller, 2017; Dai et al., 2019) and template-free (TF) (Liu et al., 2017; Karpov et al., 2019) methods. Generally, TB methods achieve high accuracy by leveraging reaction templates, which encode the molecular changes during the reaction. However, the usage of templates brings some shortcomings, such as high computation cost and incomplete rule coverage, limiting the scalability. To improve the scalability, a class of chemically inspired TF methods (Shi et al., 2020; Yan et al., 2020) (see Fig. 1) have achieved dramatical success, which decompose retrosynthesis into subproblems: i) *center identification* and ii) *synthon completion*. Center identification increases the model scalability by breaking down the target molecule into virtual synthons without utilizing templates. Synthon completion simplifies reactant generation by taking synthons as potential starting molecules, i.e., predicting residual molecules and attaching them to synthons to get reactants. Although various TF methods have been proposed, the top-$k$ retrosynthesis accuracy remains poor, especially when the reaction class is unknown. Can we find a more accurate way to predict potential reactants while keeping the scalability?

To address the aforementioned problem, we suggest combining the advantages of TB and TF approaches, and propose a novel framework namely SemiRetro. Specifically, we break a full-template

---

[1]In single-step retrosynthesis, the synthesis route length is 1, i.e., only one reaction needs to be inferred.

into several simpler semi-templates and embed them into the two-step TF framework. As many semi-templates are reduplicative, the template redundancy can be reduced while the essential chemical knowledge is still preserved to facilitate synthon completion. Moreover, we introduce a directed relational graph attention (DRGAT) layer to extract more expressive molecular features to improve center identification accuracy. Finally, we combine the center identification and synthon completion modules in a unified framework to accomplish retrosynthesis predictions.

We evaluate the effectiveness of SemiRetro on the benchmark data set USPTO-50k, and compare it with recent state-of-the-art TB and TF methods. We show that SemiRetro significantly outperforms these methods. In scalability, SemiRetro covers 96.9% of data using 150 semi-templates, while previous template-based GLN requires 11,647 templates to cover 93.3% of data. In top-1 accuracy, SemiRetro exceeds template-free G2G 3.4% (class known) and 6.4% (class unknown). Owing to the semi-template, SemiReto is more interpretable than template-free G2G and RetroXpert in synthon completion. Moreover, SemiRetro trains at least 6 times faster than G2G, RetroXpert, and GLN. All these results show that the proposed SemiRetro boosts the scalability and accuracy of deep retrosynthesis prediction.

## 2 RELATED WORK

**Template-based models**   TB methods infer reactants from the product through shared chemical transformation patterns, namely reaction templates. These templates are either hand-crafted by human experts (Hartenfeller et al., 2011; Szymkuć et al., 2016) or automatically extracted by algorithms (Coley et al., 2017a; Law et al., 2009). For a product molecule, due to the vast search space, multiple qualified templates, and non-unique matching sites for each template, it is challenging to select and apply the proper template to generate chemically feasible reactants. To handle those challenges, (Coley et al., 2017b) suggests sharing the same templates among similar products. (Segler & Waller, 2017; Baylon et al., 2019) employ neural models for template selection with molecule fingerprint as input. The state-of-the-art GLN (Dai et al., 2019) learns the joint distribution of templates and products by decomposing templates into pre-reaction and post-reaction parts and introducing logic variables to apply structure constraints. TB methods are interpretable and accurate because they embed rich chemical knowledge into the algorithm. However, the vast space of templates and incomplete coverage severely limit their scalability.

**Template-free models**   Instead of explicitly using templates, TF approaches learn chemical transformations by the model. (Liu et al., 2017; Karpov et al., 2019) solve the retrosynthesis problem with seq2seq models, e.g. Transformer (Vaswani et al., 2017), LSTM (Hochreiter & Schmidhuber, 1997), based on the SMILES representation of molecules. Despite the convenience of modeling, SMILES cannot fully utilize the inherent chemical structures and may generate invalid SMILES strings. Therefore, (Zheng et al., 2019) propose a self-corrected transformer to fix the syntax errors of candidate reactants. Recently, G2G (Shi et al., 2020), RetroXpert (Yan et al., 2020) and GraphRetro (Somnath et al., 2021) achieve state-of-the-art performance by decomposing the retrosynthesis into two sub-problems: i) center identification and ii) synthon completion, as shown in Fig. 1. Center identification increases the model scalability by breaking down the target molecule into virtual synthons without utilizing templates, among which G2G reports the highest accuracy. Synthon completion simplifies the complexity of reactant generation by taking synthons as potential starting molecules. For example, RetroXpert and G2G treat it as a SMILES or graph sequence translation problem from synthon to reactant. GraphRetro completes synthons by predicting pre-defined leaving groups, but it does not provide end-to-end open-source algorithms for attaching leaving groups and model construction. Generally, these TF methods are more scalable but perform worse than TB GLN in top-1 accuracy.

**Challenges**   Although the two-step TF framework significantly improves the algorithm's scalability, the overall accuracy is relatively low. A possible solution to this issue is to enhance submodules, i.e., center identification and synthon completion. 1) To the best of our knowledge, current GNN models only work well for center identification when the reaction class is known; Otherwise, the model performance degrades rapidly. *How to develop a more suitable model that works well with unknown classes* is the first challenge. 2) In addition, synthon completion is the major bottleneck affecting the overall accuracy. Specifically, predicting and attaching residuals for each synthon are

difficult because the residual structures could be complex, attaching residuals into synthons may violate chemical rules, and various residuals may agree with the same synthon (e.g., F, CI, Br, and I have similar chemical properties). For researchers, scalability, interpretability, and training efficiency are also important. *How to develop a more accurate, interpretable, and efficient synthon completion model while maintaining the scalability* is the second challenge.

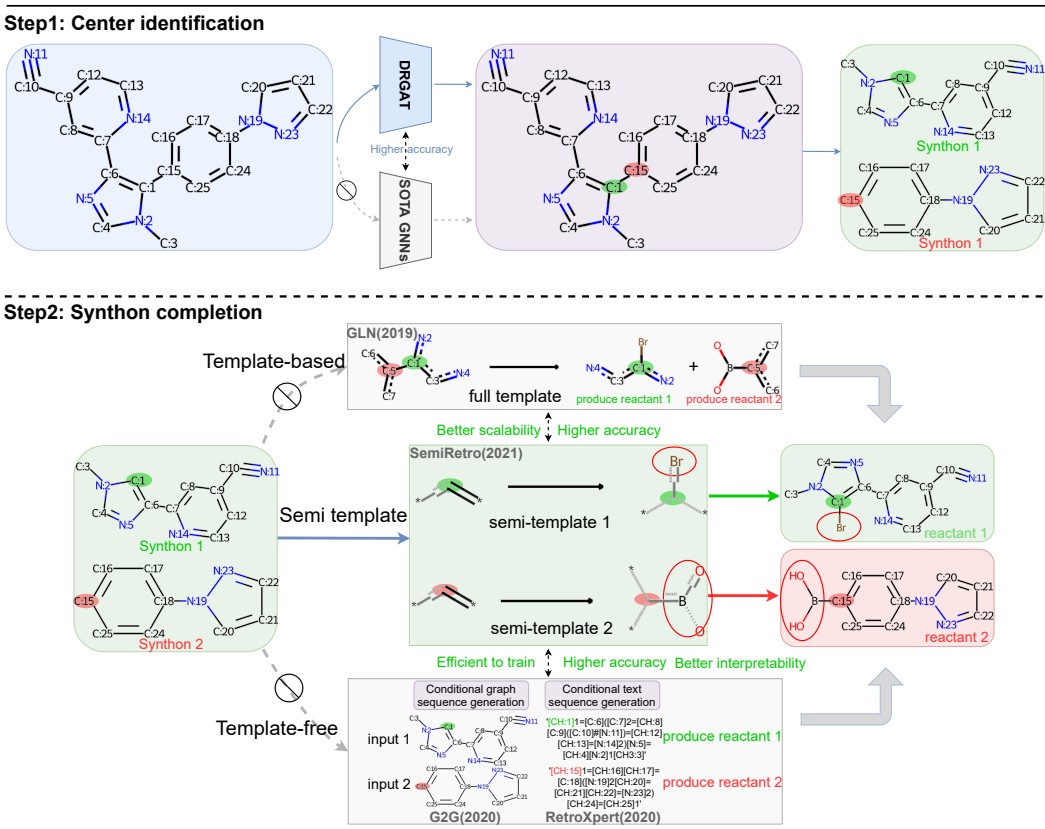

**Figure 1:** Overview of SemiRetro. We decomposite retrosynthesis into two steps: *center identification* and *synthon completion*. In step 1, we use DRGAT to extract molecule features for predicting reaction centers. By breaking product bonds in these centers, synthons can be obtained. In step 2, we use another DRGAT model to predict the semi-template for each synthon. The final reactants can be deduced from reaction centers, synthons, and semi-templates by using the residual attachment algorithm.

## 3 DEFINITION AND OVERVIEW

**Molecule representation** There are two types of dominant representations, i.e., SMILES string (Weininger, 1988) and molecular graph. SMILES is commonly used in early works (Liu et al., 2017; Schwaller et al., 2018; Zheng et al., 2019; Schwaller et al., 2019; Tetko et al., 2020) due to its simpleness. Many NLP models can be directly applied to solve related problems in an end-to-end fashion. However, these models cannot guarantee the chemical correctness of the output molecules because they ignore structure information to some extent. Similar to recent breakthroughs (Dai et al., 2019; Shi et al., 2020; Yan et al., 2020; Somnath et al., 2021), we take the molecule as a labeled graph $\mathcal{G}(A, X, E)$, where $A$, $X$ and $E$ are adjacency matrix, atom features and bond features, seeing Table. 1. Under the graph framework, we can effectively apply chemical constraints to ensure the validity of output molecules. Besides, graph-based methods are more controllable and interpretable than SMILES-based ones.

**Problem definition** Retrosynthesis aims to infer the set of reactants $\{\mathcal{R}_i\}_{i=1}^N$ that can generate the product $\mathcal{P}$. Formally, that is to learn a mapping function $\mathcal{F}_\theta$:

$$\mathcal{F}_\theta : \mathcal{P} \mapsto \{\mathcal{R}_i\}_{i=1}^{N_1}. \tag{1}$$

Considering the unknown by-products, the law of conservation of atoms no longer holds here, which makes the problem quite challenging because the algorithm needs to generate new atoms and bonds to get potential reactants automatically.

**Overview** As shown in Fig. 1, we adopt the two-step TF framework due to its scalability and effectiveness. Our method is distinguished from previous works in two folds: 1) We propose a relational graph attention (DRGAT) layer to improve the center identification performance; 2) We use semi-templates to facilitate synthon completion, which significantly reduces the problem complexity.

**Table 1:** Commonly used symbols

| Symbol | Description |
|--------|-------------|
| $\mathcal{G}(A, X, E)$ | Molecular graph with adjacency matrix $A$, atom features $X$ and bond features $E$. |
| $A$ | $A \in \{0,1\}^{n,n}$ with the number of atoms $n$. $a_{i,j} = 1$ indicates that there is a bond between atom $i$ and $j$ and vice versa. |
| $X$ | $X \in \mathbb{R}^{n,d}$, $d$ is the dimension of atom features. $\boldsymbol{x}_i$ is feature vector of atom $i$. |
| $E$ | $E \in \mathbb{R}^{m,b}$, $b$ is bond features dimension and $m$ is the number of bonds. |
| $\boldsymbol{e}_{i,j}$ | feature vector of bond $(i, j)$. |
| $\mathcal{R}_i, \mathcal{S}_j, \mathcal{P}$ | a molecular graph of the $i$-th reactant, the $j$-th synthon and the product. |
| $c_i$ | $c_i \in \{0,1\}$, indicating whether atom $i$ is the reaction center or not. |
| $c_{i,j}$ | $c_{i,j} \in \{0,1\}$, indicating whether bond $(i, j)$ is the reaction center or not. |

## 4 METHODOLOGY

### 4.1 CENTER IDENTIFICATION

Center identification plays a vital role in the two-step retrosynthesis because errors caused by this step directly lead to the final failures. Previous works have limitations, e.g., RetroXpert (Yan et al., 2020) provides incomplete prediction without considering atom centers, and G2G (Shi et al., 2020) performs poorly when the reaction class is unknown. How to obtain comprehensive and accurate center identification results is still worth exploring.

**Reaction centers** We consider both atom centers and bond centers in the product molecule. As shown in Fig. 2, from the product to its corresponding reactants, either some atoms add residuals by dehydrogenation without breaking the product structure (case 1), or some bonds are broken to allow new residues to attach (case 2). Both these atoms and bonds are called reaction centers.

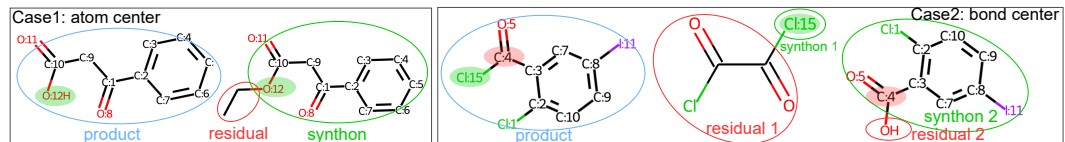

**Figure 2:** Reaction centers. Products, reactants, and residuals are circled in blue, green, and red, respectively. We label atoms in reaction centers with solid circles. In case 1, the centered atom adds residual by removing hydrogen atoms without changing the product structure. In case 2, the centered bond in the product is broken to generate synthons, to which new residuals are attached.

**Directed relational GAT** Commonly used graph neural networks (Defferrard et al., 2016; Kipf & Welling, 2016; Veličković et al., 2017) mainly focus on 0 and 1 edges, ignoring edge direction and multiple types, thus failing to capture expressive molecular features. As to molecules, different bonds represent different interatomic interactions, resulting in a multi-relational graph. Meanwhile, atoms at the end of the same bond may gain or lose electrons differently, leading to directionality. Considering these factors, we propose a directed relational graph attention (DRGAT) layer based on the general information propagation framework Zheng et al. (2021), as shown in Fig. 3. During message passing, DRGAT extracts source and destination node's features via independent MLPs to consider the bond direction and use the multi-head edge controlled attention mechanism to consider the multi-relational properties. We add shortcut connections from the input to the output in each layer and concatenate hidden representations of all layers to form the final node representation.

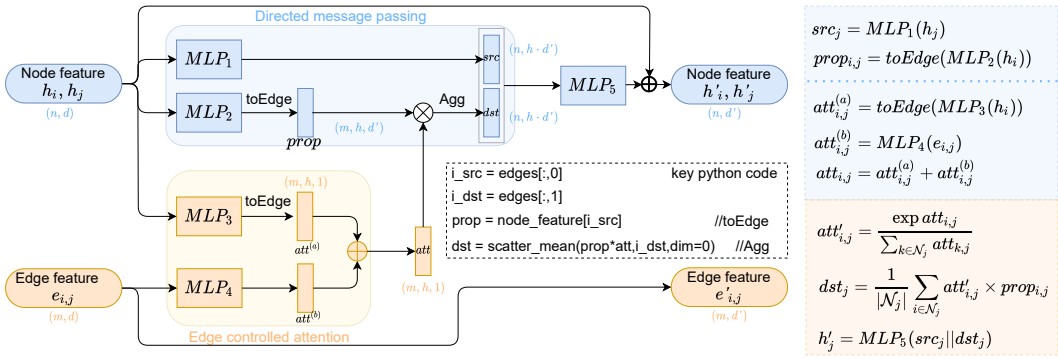

**Figure 3:** DRGAT: Directed Relational GAT. DRGAT contains two submodules: directed message passing (DMP) and edge-controlled attention (ECA). DMP uses different MLP to learn features of the source (src) and target (dst) atoms during message passing. ECA utilizes both atom features and bond features to learn the attention weights.

**Labeling and learning reaction centers**  We use the same labeling algorithm as G2G to identify ground truth reaction centers, where the core idea is comparing each pair of atoms in the product $\mathcal{P}$ with that in a reactant $\mathcal{R}_i$. We denote the atom center as $c_i \in \{0, 1\}$ and bond center as $c_{i,j} \in \{0, 1\}$ in the product $\mathcal{P}$. During the learning process, atoms features $\{h_i\}_{i=1}^{|\mathcal{P}|}$ are learned from the product $\mathcal{P}$ by applying stacked DRGAT, and the input bond features are $\{e_{i,j}|a_{i,j} = 1\}$. Then, we get the representations of atom $i$ and bond $(i, j)$ as

$$\underbrace{\hat{h}_i = h_i||\text{Readout}(\{h_s\}_{s=1}^{|\mathcal{P}|})}_{\text{atom representation}} \quad \text{and} \quad \underbrace{\hat{h}_{i,j} = e_{ij}||h_i||h_j||\text{Readout}(\{h_s\}_{s=1}^{|\mathcal{P}|})}_{\text{bond representation}}, \quad (2)$$

where we use average Readout operation, and $||$ is vector concatenation operation. Further, we predict the atom center probability $p_i$ and bond center probability $p_{i,j}$ via MLPs:

$$p_i = \text{MLP}_6(\hat{h}_i) \quad \text{and} \quad p_{i,j} = \text{MLP}_7(\hat{h}_{i,j}). \quad (3)$$

Finally, center identification can be reduced to a binary classification, whose loss function is:

$$\mathcal{L}_1 = \sum_{\mathcal{P}} (\underbrace{\sum_i c_i \log p_i + (1 - c_i) \log p_i}_{\text{atom center}} + \underbrace{\sum_{i,j} c_{i,j} \log p_{i,j} + (1 - c_{i,j}) \log p_{i,j}}_{\text{bond center}}). \quad (4)$$

In summary, we propose a directed relational graph attention (DRGAT) layer to learn expressive atom and bond features for accurate center identification prediction. We consider both atom center and bond center to provide comprehensive results. In section. 5.2, we show that our method can achieve state-of-the-art accuracy, especially with reaction class unknown.

## 4.2 SYNTHON COMPLETION

Synthon completion is the main bottleneck of two-step TF retrosynthesis, which is responsible for predicting and attaching residuals for each synthon. This task is challenging because the residual structures could be complex to predict, attaching residuals into synthons may violate chemical rules, and various residuals may agree with the same synthon. Because of these complexities, previous synthon completion approaches are usually inaccurate, unexplainable, and cumbersome. Introducing the necessary chemical knowledge to improve interpretability and accuracy can be a promising solution. However, how to provide attractive scalability and training efficiency is a new challenge.

**Semi-templates** The semi-template used in this paper is the local reaction pattern of each synthon, as shown in Fig. 4. Different from GraphRetro (Somnath et al., 2021), our semi-template contains not only residuals but also synthon's local structure. Similar to the work of forward reaction prediction (Segler & Waller, 2016), semi-template splits a binary reaction into two half reactions. Notably, we use dummy atom ∗ to represent possible synthon atoms that match the semi-template, significantly reducing redundancy. We extract semi-template from each synthon-reactant pair by removing reactant atoms that have exact matches in the synthon. There are two interesting observations: 1) Top-150 semi-templates cover 96.9% samples; 2) Reactants can be deterministically generated from semi-templates and synthons (introduced later). Based on these observations, synthon completion can be further simplified as a classification problem. In other words, we need to predict the semi-template type for each synthon, and the total number of classes is 150+1. The first 150 classes are top-150 semi-templates, and the 151st class indicates uncovered classes.

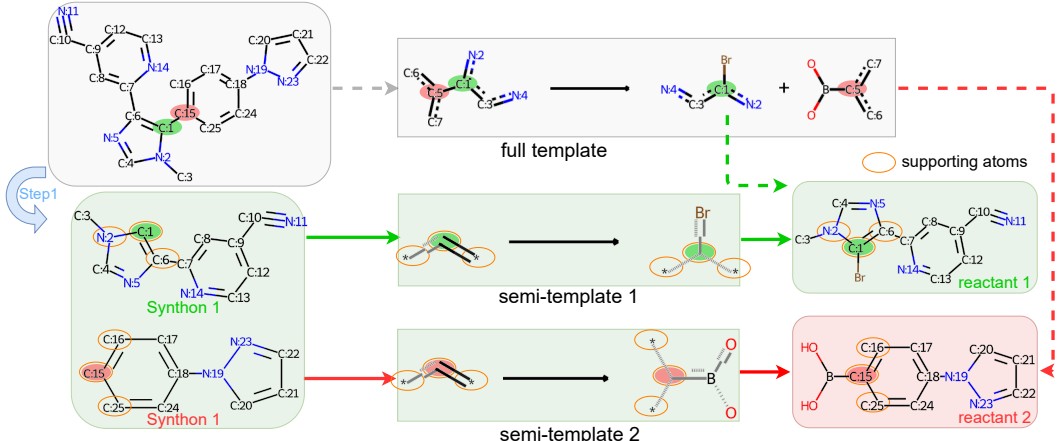

**Figure 4:** Semi-template vs. full-template. Semi-template is the local reaction template of each synthon. A full-template can be decomposed into several simpler semi-templates.

**Learning semi-templates** For each synthon $\mathcal{S}_j$, denote its semi-template label as $t_j, 1 \leq t_j \leq 151$. We use stacked DRGATs to extract atom features $\{\boldsymbol{h}_i\}_{i=1}^{|\mathcal{S}_j|}$ from $\mathcal{S}_j$. In each synthon, atoms that exactly match the semi-template are called supporting atoms, seeing circled atoms in Fig. 4. Given the supporting atom set $\{i_0, i_1, \cdots\}$, the semi-template representation is:

$$\hat{\boldsymbol{h}}_j = \text{Readout}(\{\boldsymbol{h}_i\}_{i=1}^{|\mathcal{S}_j|})||\text{Readout}(\{\boldsymbol{h}_k|k \in \{i_0, i_1, \cdots\}\}). \tag{5}$$

Based on this representation, we can get the predicted semi-template $\hat{t}_j$ as:

$$\hat{t}_j = \underset{1 \leq c \leq 151}{\arg\max} \, \tilde{p}_{j,c}; \quad \tilde{\boldsymbol{p}}_j = \text{Softmax}(\text{MLP}_8(\hat{\boldsymbol{h}}_j)). \tag{6}$$

Denote $\mathbb{1}_{\{c\}}(\cdot)$ as the indicator function, the cross-entropy loss used for training is:

$$\mathcal{L}_2 = - \sum_{j \in \{1,2,\cdots,|\mathcal{S}|\}} \sum_{1 \leq c \leq 151} \mathbb{1}_{\{c\}}(t_j) \log(\tilde{p}_{j,c}). \tag{7}$$

**Applying semi-templates** Once reaction centers, synthons, and corresponding semi-templates are known, we can deduce reactants with almost 100% accuracy. This is not a theoretical claim; We provide a practical residual attachment algorithm in the appendix and open source code on GitHub.

In summary, we suggest using the semi-templates to improve synthon completion performance. Firstly, reducing this complex task to a classification problem helps promote training efficiency and accuracy. Secondly, the high coverage of semi-templates significantly enhanced the scalability of TB methods. Thirdly, the deterministic residual attachment algorithm improves interpretability and accuracy. In section. 5.3, we will show the effectiveness of the proposed method.

## 5 EXPERIMENTS

As mentioned earlier, the main contributions of this paper are proposing a DRGAT layer for central identification and suggesting to use semi-template for synthon completion. The effectiveness of the proposed method is evaluated by systematic experiments, which focus on answering these questions:

- **Q1:** For center identification (CI), how much performance gain can be obtained from DRGAT?
- **Q2:** For synthon completion (SC), can semi-templates reduce template redundancy and improve the synthon completion performance? And why?
- **Q3:** For retrosynthesis, how do we integrate CI and SC models into a unified retrosynthesis framework? Can SemiRetro outperform existing template-based and template-free methods?

### 5.1 BASIC SETTING

**Data**   We evaluate SemiRetro on the widely used benchmark dataset USPTO-50k (Schneider et al., 2016) to show its effectiveness. USPTO-50k contains 50k atom-mapped reactions with 10 reaction types. Following (Dai et al., 2019; Shi et al., 2020; Yan et al., 2020), the training/validation/test splits is 8:1:1. As mentioned in previous works, the USPTO-50k dataset contains a shortcut in that the product atom with atom-mapping "1" is part of the reaction center in 75% of the cases. In our graph-based methods, this shortcut will not be introduced by forbidding atom position encoding.

**Baselines**   Template-based GLN (Dai et al., 2019), template-free G2G (Shi et al., 2020) and RetroXpert (Yan et al., 2020) are primary baselines, which not only achieve state-of-the-art performance, but also provide open-source PyTorch code that allows us to verify their effectiveness. To show broad superiority, we also comapre SemiRetro with other baselines, incuding RetroSim (Coley et al., 2017b), NeuralSym (Segler & Waller, 2017), SCROP (Zheng et al., 2019), LV-Transformer (Chen et al., 2019), GraphRetro (Somnath et al., 2021), MEGAN (Sacha et al., 2021), and MHNreact (Seidl et al., 2021). As the retrosynthesis task is quite complex, subtle implementation differences or mistakes may cause critical performance fluctuations. We prefer comparing SemiReto with open-source methods whose results are more reliable. Energy-based approaches (Sun et al., 2020) are ignored because they are more like plug-and-play training strategies and result filters, focusing on enhancing existing models. For simplicity, we leave the use of energy function in the future and concentrate on comparing original retrosynthesis models.

**Metrics**   This paper uses consistent metrics derived from previous literature for both submodule and overall performance. 1). *Center identification*: We report the accuracy of breaking input product into synthons. 2). *Synthon completion*: We present the accuracy of generating reactants from ground truth input synthons. When a product has multiple synthons, the final prediction is correct if and only if all reactants are correct. 3). *Retrosynthesis*: The metric is similar to that of synthon completion, except that the input synthons are also predicted by center identification. In other words, the retrosynthesis is correct if and only if both center identification and synthon completion are correct. Since there may be multiple valid routes for synthesizing a product, we report top-$k$ accuracy.

**Implementation details**   Thanks to the elegant implementation of G2G (Shi et al., 2020), we can develop our SemiRetro in a unified PyTorch framework (Paszke et al., 2019), namely TorchDrug. We use the same data preprocessing algorithm and code framework as G2G in center identification, ensuring the only difference is the GNN used for feature extraction. Besides, we use the open-source cheminformatics software RDkit (Landrum, 2016) to preprocess molecules and SMILES strings. The graph feature extractor consists of 6 stacked DRGAT, with the embedding size 256 for each layer. We train the proposed model for 30 and 100 epochs in center identification and synthon completion with batch size 64 on a single NVIDIA V100 GPU. The training costs of different methods can be found in the appendix. We run all experiments three times and report the means of their performance in default. To avoid the label leakage Yan et al. (2020); Somnath et al. (2021), we ignore atom mapping numbers during the training and evaluation phases.

### 5.2 CENTER IDENTIFICATION (**Q1**)

**Objective and setting**   This section studies how much center identification performance gain can be obtained from the proposed DRGAT, especially when the reaction class is unknown. The data preprocessing, training, and evaluating process is similar to G2G. The primary difference between our SemiRetro and G2G is the graph feature extractor, where we use DRGAT while G2G uses RGCN

(Schlichtkrull et al., 2018). We trained our model up to 30 epochs, which occupied about 2940 MB of GPU memory, where the batch size is 64, and the learning rate is 1e-5.

**Results and analysis** 1) **Highest accuracy**: As shown in 2, the proposed SemiRetro outperforms baselines on all the cases with different $k$. For example, SemiRetro achieves the highest top-1, top2, top-3, top-5 accuracy on both reaction class known and unknown settings. 2) **Better potential**: Since the possible synthesis routes toward a product may be multiple, the top-$k$ accuracy ($k > 1$) is important, and the performance gain of SemiRetro rises as $k$ increases, indicating the better potential. In particular, SemiRetro achieves nearly perfect top-5 accuracy on the setting of reaction class known (acc = 99.4%) and unknown (acc = 98.4%). 3) **Better adaptability**: In a more general and complex case, where the reaction class is unknown, SemiRetro can significantly exceed SOTA methods. For example, SemiRetro outperforms competitors by at least 10% on the top-3 and top-5 accuracy. These results show that the center identification performance has been dramatically improved using DRGAT, which is a good first step towards accurate retrosynthesis prediction.

| | Top-$k$ center identification accuracy | | | | | | | |
| | Reaction class known | | | | Reaction class unknown | | | |
| $k=$ | 1 | 2 | 3 | 5 | 1 | 2 | 3 | 5 |
|---|---|---|---|---|---|---|---|---|
| RetroXpert (Yan et al., 2020) | 86.0 | – | – | – | 64.9 | – | – | – |
| GraphRetro (Somnath et al., 2021) | 84.6 | 92.2 | 93.7 | 94.5 | 70.8 | 85.1 | 89.5 | 92.7 |
| G2G (Shi et al., 2020) | 90.2 | 94.5 | 94.9 | 95.0 | 75.8 | 83.9 | 85.3 | 85.6 |
| SemiRetro (our) | **90.9** | **97.2** | **98.5** | **99.4** | **80.5** | **92.8** | **96.0** | **98.4** |
| Improvement | +0.7 | +2.7 | +3.6 | +4.4 | +4.7 | +8.9 | +10.7 | +12.8 |

**Table 2:** Top-$k$ center identification accuracy. The best and sub-optimum results are highlighted in bold and underline.

## 5.3 SYNTHON COMPLETION (**Q2**)

**Objective and setting** This section reveals the effectiveness of using semi-template in three folds: 1) reducing the template redundancy, 2) improving the accuracy, and 3) promoting interpretability and training efficiency. Firstly, we count the full-templates of GLN and semi-templates introduced in this paper. We visualize the distribution and coverage of top-$k$ templates for analyzing the redundancy. Secondly, we present the accuracy of synthon completion with ground truth synthon inputs. The final reactants are obtained by predicting the semi-templates and applying the residual attachment algorithm. Thirdly, we compare the interpretability and training efficiency of different methods in short. We trained our model up to 100 epochs, which occupied about 2320 MB of GPU memory, where the batch size is 64, and the learning rate is 1e-4.

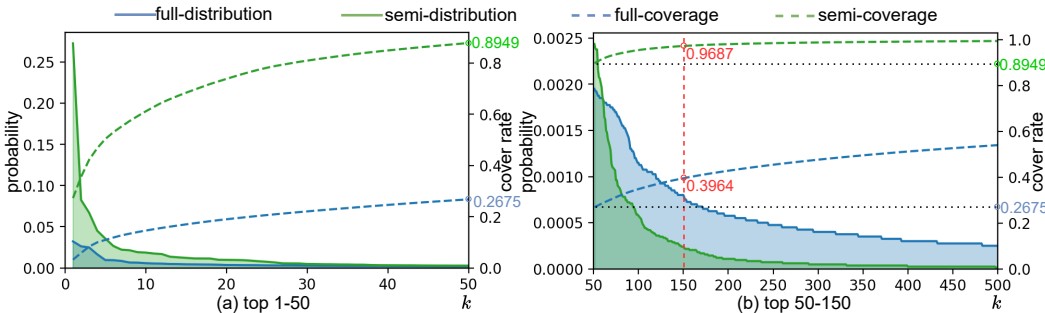

**Figure 5:** SemiRetro reduces the template redundancy.

| | top-$k$ synthon completion accuracy | | | | | | | |
| | Reaction class known | | | | Reaction class unknown | | | |
| $k=$ | 1 | 3 | 5 | 10 | 1 | 3 | 5 | 10 |
|---|---|---|---|---|---|---|---|---|
| G2G (Shi et al., 2020) | 66.8 | 87.2 | **91.5** | **93.9** | 61.1 | 81.5 | 86.7 | 90.0 |
| SemiRetro | **70.2** | **87.7** | **91.5** | 93.2 | **66.3** | **85.6** | **90.0** | **92.7** |
| Improvement | +3.4 | +0.5 | +0.0 | – | +5.2 | +4.1 | +3.3 | +2.7 |

**Table 3:** Top-$k$ synthon completion accuracy.

**Results and analysis** (1) **Reduce redundancy**: In Fig. 5, we show the distribution and coverage of top-$k$ full-templates and semi-templates, where the former distribution is sharper than the latter, indicating a higher top-$k$ coverage. For example, the top-150 semi-templates cover the case of 96.87%, while the full-templates only cover 39.64%. Using semi-templates can reduce 11,647 full-templates into 150 semi-templates and increase the cover rate from 93.3% to 96.87%. (2) **Higher accuracy** As shown in Table. 3, SemiRetro outperforms G2G in most cases. This improvement comes from two parts: Firstly, semi-templates reduce the difficulty of predicting residual structures; Secondly, semi-templates eliminate the problem of residual attachment by using a deterministic algorithm. When the reaction class is unknown, the improvement is more dramatic. (3) **More interpretable and efficient** By using semi-templates, the residual attachment process is controllable and interpretable. In addition, our model can be trained at least 8 times faster than previous synthon completion models such as GLN, G2G, and RetroXpert, seeing the appendix for details.

### 5.4 RETROSYNTHESIS (Q3)

**Objective and setting** We explain how to combine center identification and synthon completion to provide end-to-end retrosynthesis predictions. We use a probability tree to search the top-$k$ results, seeing Fig. 6, where the probability product of two-step predictions is used to rank these results.

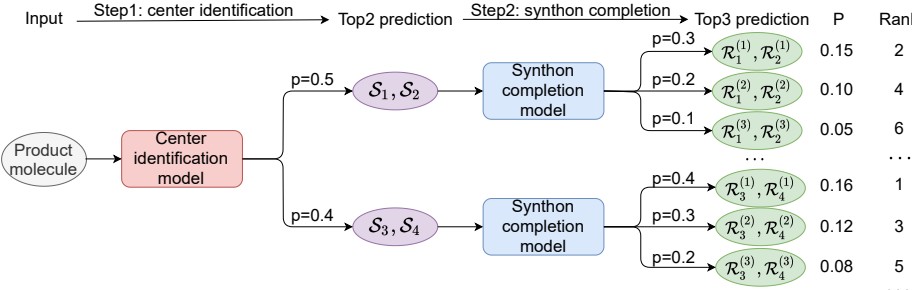

**Figure 6:** The retrosynthesis example: combining top-2 CI and top-3 SC to obtain top-6 retrosynthesis results. Note that $\mathcal{S}_i$ indicates the $i$-th synthon, and $\mathcal{R}_i^{(j)}$ is the $j$-th predicted reactant of $\mathcal{S}_i$.

| | $k=$ | top-$k$ accuracy | | | | | | | |
| | | Reaction class known | | | | Reaction class unknown | | | |
| | | 1 | 3 | 5 | 10 | 1 | 3 | 5 | 10 |
|---|---|---|---|---|---|---|---|---|---|
| TB | RetroSim (Coley et al., 2017b) | 52.9 | 73.8 | 81.2 | 88.1 | 37.3 | 54.7 | 63.3 | 74.1 |
| | NeuralSym (Segler & Waller, 2017) | 55.3 | 76.0 | 81.4 | 85.1 | 44.4 | 65.3 | 72.4 | 78.9 |
| | GLN (Dai et al., 2019) | 64.2 | 79.1 | 85.2 | 90.0 | 52.5 | 69.0 | 75.6 | 83.7 |
| TF | SCROP (Zheng et al., 2019) | 59.0 | 74.8 | 78.1 | 81.1 | 43.7 | 60.0 | 65.2 | 68.7 |
| | LV-Transformer (Chen et al., 2019) | – | – | – | – | 40.5 | 65.1 | 72.8 | 79.4 |
| | G2G (Shi et al., 2020) | 61.0 | 81.3 | 86.0 | 88.7 | 48.9 | 67.6 | 72.5 | 75.5 |
| | RetroXpert (Yan et al., 2020) | 62.1 | 75.8 | 78.5 | 80.9 | 50.4 | 61.1 | 62.3 | 63.4 |
| | GraphRetro (Somnath et al., 2021) | 63.9 | 81.5 | 85.2 | 88.1 | 53.7 | 68.3 | 72.2 | 75.5 |
| | MEGAN (Sacha et al., 2021) | 60.7 | 82.0 | 87.5 | **91.6** | 48.1 | 70.7 | 78.4 | 86.1 |
| | MHNreact (Seidl et al., 2021) | – | – | – | – | 50.5 | 73.9 | 81.0 | **87.9** |
| Our | SemiRetro | **64.4** | **83.7** | **87.6** | 90.2 | **55.3** | **76.5** | **81.4** | 85.4 |
| | Improvement to GLN | +0.2 | +4.6 | +2.4 | +0.2 | +2.8 | +7.5 | +5.8 | +1.7 |
| | Improvement to G2G | +3.4 | +2.4 | +1.6 | +1.5 | +4.9 | +8.9 | +8.6 | +6.0 |

**Table 4:** Overall performance. The best and sub-optimum results are highlighted in bold and underline. Only open source works participate in rigorous comparisons, such as GLN, G2G, and RetroXpert.

**Results and analysis** (1) **Higher accuracy**: SemiRetro achieves the highest accuracy in most settings, seeing Table. 4. As to previous works with open source code, template-free G2G and RetroXpert are more scalable than template-based GLN while sacrificing the top-1 accuracy. We use semi-template to reduce the template redundancy and improve the accuracy simultaneously. (2) **Better adaptability** We observe the performance gain of SemiRetro increases when the reaction class is unknown, suggesting our method can work better in more general and complex cases.

## 6 CONCLUSION

We propose SemiRetro for retrosynthesis prediction, which achieves better accuracy and attractive scalability than previous methods. Specifically, the DRGAT achieves the highest center identification accuracy. The semi-template improves both the accuracy and scalability of synthon completion. Moreover, SemiRetro has favorable interpretability and training efficiency. We hope this work will promote the development of deep retrosynthesis prediction.

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

## A APPENDIX

**Center identification** We show top-2 center identification predictions in Fig. 7, where synthons are obtained from breaking edge centers for downstream synthon completion. We present the probability of each prediction where the total probability of top-2 predictions exceeds 98%, indicating strong inductive confidence. Since the top-2 predictions are accurate enough, seeing Table. 2, we use them for synthon completion.

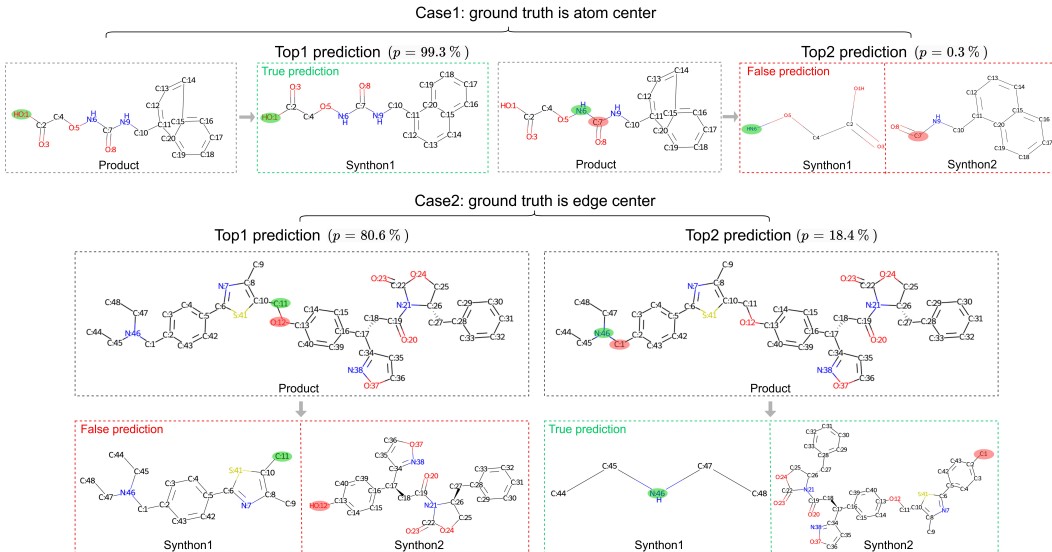

**Figure 7:** Visualize results of center identification. Case1: the ground truth is atom center, and the top-1 prediction is correct with the probability 99.3%. Case2: The ground truth is edge center, and the top-2 prediction is correct with the probability 18.4%.

**Synthon completion** In Fig. 8, we present the process of predicting multiple reactants of the same product. This process provides an end-to-end view of synthon completion, containing semi-template prediction, top-$k$ results search, and semi-template application. By default, we choose the top-5 synthon completion results for each center identification output as part of the final top-10 retrosynthesis results.

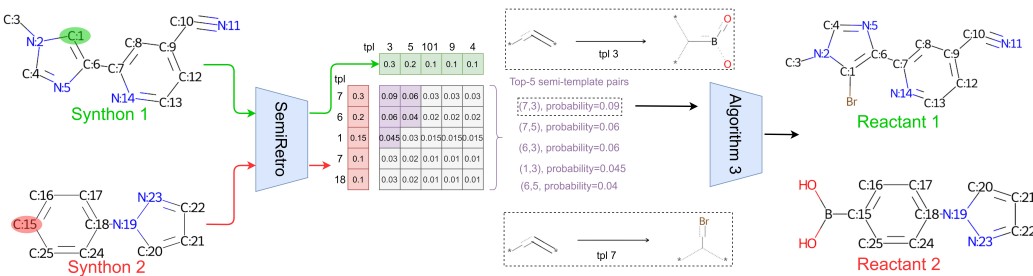

**Figure 8:** The overall pipeline of synthon completion. The input synthons are the outputs of the center identification module, coming from the same product molecule. We get the top-5 semi-template predictions and their probabilities of each synthon using SemiRetro (synthon completion network), then generate the joint distribution of semi-templates. We choose the top-5 predictions from this joint distribution and apply the residual attachment algorithm (introduced later) to get the final reactants.

**Table 5:** Residual attachment algorithm. For easy and quick understanding, we demonstrate the core idea by visual samples. The detailed implementation can be found in the open-source code.

**Input**: Synthon, reaction center, and semi-template

**Output**: Reactant obtained by applying the semi-template on the synthon

Step1: map atoms within the template

Step2: map bonds within the template

Step3: match the left template with the synthon
Constraint: the reaction center must be within the matching area

Step4: Attach the right template into the synthon

remove the left template     add the right template     final result

**Platform**   The platform for our experiment is ubuntu 18.04, with a Intel® Xeon® Gold 6240R Processor and 256GB memory. We use a single NVIDIA V100 to train models, where the CUDA version is 10.2.

| | Retrosynthesis | Center identification | | | Synthon completion | | |
|---|---|---|---|---|---|---|---|
| | GLN | RetroXpert | G2G | SemiRetro | RetroXpert | G2G | SemiRetro |
| time/epoch | 785s | 440s | 58s | **48s** | 330s | 322s | **40s** |
| GPU memory/sample | 274.7MB | 55.7MB | 46.1MB | **45.9MB** | 147.7MB | 65.7MB | **36.3MB** |
| total epochs | 50 | 80 | 100 | 30 | 300 | 100 | 100 |

**Table 6:** The training costs of different methods. We run the open-source code of these methods on the same platform, reporting the training time per epoch and occupied GPU memory per sample. We also show the total training epochs mentioned in their paper (preferred) or code. If the author reports training steps, we calculate $\text{epochs}_{\text{total}} = \text{steps}_{\text{total}}/\text{steps}_{\text{interval}}$.

**Important details**   We follow the setting of G2Gs, which provides open-source code on https://github.com/DeepGraphLearning/torchdrug/. G2Gs use different atom features in their open-source code for center identification and synthon completion. In this paper, we use the same atom features of G2Gs. We have also tried to combine all these atom features and use the same set of features in center identification and synthon completion models. The combined atom features do not make a significant difference.

| Name | Description |
|---|---|
| Atom type | Type of atom (ex. C, N, O), by atomic number |
| # Hs | one-hot embedding for the total number of Hs (explicit and implicit) on the atom |
| Degree | one-hot embedding for the degree of the atom in the molecule including Hs |
| Valence | one-hot embedding for the total valence (explicit + implicit) of the atom |
| Aromaticity | Whether this atom is part of an aromatic system. |
| Ring | whether the atom is in a ring |

**Table 7:** Atom features for center identification.

| Name | Description |
|---|---|
| Bond type | one-hot embedding for the type of the bond |
| Bond direction | one-hot embedding for the direction of the bond |
| Stereo | one-hot embedding for the stereo configuration of the bond |
| Conjugation | whether the bond is considered to be conjugated |
| Bond length | the length of the bond |

**Table 8:** Bond features for center identification.

| Name | Description |
|---|---|
| Atom type | Type of atom (ex. C, N, O), by atomic number |
| # Hs | one-hot embedding for the total number of Hs (explicit and implicit) on the atom |
| Degree | one-hot embedding for the degree of the atom in the molecule including Hs |
| Ring | whether the atom is in a ring |
| Ring 3 | whether the atom is in a ring of size 3 |
| Ring 4 | whether the atom is in a ring of size 4 |
| Ring 5 | whether the atom is in a ring of size 5 |
| Ring 6 | whether the atom is in a ring of size 6 |
| Ring 6+ | whether the atom is in a ring of size larger than 6 |

**Table 9:** Atom features for synthon completion. Note that synthon completion do not use bond features.

