# OpenReview forum: "SemiRetro: Semi-template framework boosts deep retrosynthesis prediction"
_ICLR.cc/2022/Conference — ICLR 2022 Submitted_

### Official Review · Reviewer_83gD · 2021-10-28

**Correctness:** 4
**Technical Novelty And Significance:** 3
**Empirical Novelty And Significance:** 3
**Recommendation:** 5
**Confidence:** 3

**Main Review:**

(Sorry I re-organize the review to meet the guideline. No changes in the contents/opinions from the initial submission.)

### points

(+): a new variant of retrosynthesis problem using a local-structure aware semi-template

(++): good numerical performances confirmed in the sub-problems and the main retrosynthesis

(-): insufficient explanations of DRGAT

(-): possible missing reference and comparison

(-); visibility of figures are generally low

### comments

I like the idea of this paper, and enjoy reading the manuscript.

In my understanding, the main stream of the proposed model follows the typical template-free retrosynthesis.
In the current TF approaches, the center identification sub-problem affects a lot in the final retrosynthesis problem.
The proposed DRGAT seems performing great in this sub-problem (Sec. 5.2).
However, the explanation of the DRGAT model only appears in the Figure 3. I expect the revised manuscript add a few sentences/equations to convey the main idea of the model.

The semi-template model for synthon completion is quite interesting.
It seems for me a good alternative of the GraphRetro's residual prediction approach.
Question: Why GraphRetro and RetroXpert are not studied in the Sec. 5.3? In terms of the synthon completion, GraphRetro's residual modeling may perform as good as the proposed semi-template.

I found that one of the important previous work is missing from the discussions and the experiments; energy-based models.
Please discuss the relationship with the proposed model, and if it is appropriate, compare in experiments.

@misc{
sun2021energybased,
title={Energy-based View of Retrosynthesis},
author={Ruoxi Sun and Hanjun Dai and Li Li and Steven Kearnes and Bo Dai},
year={2021},
url={https://openreview.net/forum?id=0Hj3tFCSjUd}
}


Readability: In general, figures are low-visibility. Especially, atoms, bonds, molecules, equations are not displayed well on screens.
I hope these figures are improved in the camera ready as much as possible. For example, thick/strong lines and fonts, larger symbols, and so on.

**Summary Of The Paper:**

This paper studies a retrosynthesis problem of chemical reactions.
This paper proposes a new variant of retrosynthesis models, with mainly two new components: DRGAT for powerful embedding computation and the semi-templates for accurate synthon completions.
Especially, the propose semi-template takes the local (neighboring) bonding structure into consideration. This significantly reduces the redundancy of the templates, allowing connecting the template-based retrosynthesis (less scalable but powerful) and the template-free retrosynthesis (scalable but less powerful).
The experimental validations show the efficacy of the new components for each subtask, and the final retrosynthesis performances exceeds the existing retrosynthesis models.

**Summary Of The Review:**

It is great to see the stead improvement of the retrosynthesis.
The semi-template idea sounds reasonable and good to me.
The explanations on the DRGAT is clearly not enough, and in general, figures are hard to see on the screen.
My positive score is with the expectation that the insufficient explanations and low-visibility figures are properly addressed in revisions or the camera-ready.

=============================================================

After author feedback;
Thank you again authors for the efforts to revise the manuscript and discussions.
One of the reviewers raise the concern about the leakage problem of the USPTO dataset, and I agree the problem is important.
Some strong competitors are also introduced by other reviewers.
As a result, we observe a few but important updates that are not reflected in the revised manuscript.
I still believe the proposed semi-template approach is new and valuable to the community, but this time I cannot recommend this manuscript for acceptance. I expect the authors to come up with a manuscript including all new updates in near-future conferences.

---

> ### Author Response · Authors · 2021-11-16
> **Reply to R4**
>
> **Reply to R4**
> Thanks for your comments, we are glad that you enjoy reading this paper.
>
>
> > **Q1** The explanation of the DRGAT model only appears in the Figure 3. I expect the revised manuscript add a few sentences/equations to convey the main idea of the model.
>
> **A1** Thanks for your constructive suggestion. In our revised version, we have added a detailed description in Sec. 4.1. During message passing, DRGAT extracts source and destination node's features via independent MLPs to consider the bond direction and use the multi-head edge controlled attention mechanism to consider the multi-relational properties. We add shortcut connections from the input to the output in each layer, concatenate hidden representations of all layers to form the final node representation.
>
>
> > **Q2** Why GraphRetro and RetroXpert are not studied in the Sec. 5.3?
>
> **A2** GraphRetro \citep{somnath2020learning} is difficult to reproduce, especially when open-source code is unavailable (its data preprocessing and residual attaching algorithm are unknown), so we abandon this baseline. RetroXpert \cite{NEURIPS2020_819f46e5} leaks atom mapping information, we doubt this setting is not the same as ours so that we give it up in comparison. We try to reproduce the revised version of RetroXpert, but do not get reasonable results following the official instructions, seeing \url{https://github.com/uta-smile/RetroXpert}. As G2G provides open-source code, we can conviently implement the same chemical features and code framework to ensure the fairness of experimental settings.
>
>
> > **Q3** I found that one of the important previous work is missing from the discussions and the experiments; energy-based models. Please discuss the relationship with the proposed model, and if it is appropriate, compare in experiments.
>
> **A3** Thanks for your kind suggestion. In our revised version, the specified discussion have been added in Sec. 5.1. The energy-based approaches \citep{sun2020energy} are ignored because it is more like a plug-and-play training strategy and result filter, focusing on enhancing existing models. For simplicity, we leave the use of energy function in the future and concentrate on comparing original retrosynthesis models.
>
> > **Q4** Readability: In general, figures are low-visibility. Especially, atoms, bonds, molecules, equations are not displayed well on screens. I hope these figures are improved in the camera ready as much as possible. For example, thick/strong lines and fonts, larger symbols, and so on.
>
> **A4** Thanks for your careful suggestion. We have updated all the figures in our paper to improve the readability.

---

> > ### Comment · Reviewer_83gD · 2021-11-22
> > **Thanks!**
> >
> > Thank you for the clarifications. I think I understand the difficulty of comparisons with GraphRetro and RetroXpert.
> > I also find the revised manuscript has better figures, easier to read.
> > The additional experiments with several GNN workhorses are also quite interesting.
> >
> > Other reviewers raise a novelty issue and an afraid of atom map leaks.
> > Also some missing references of the latest works, where I also overlooked.
> > I will fix my final evaluations after reviewer discussions with these issues.

---

> > > ### Author Response · Authors · 2021-11-23
> > > **Reply to Reviewer 83gD**
> > >
> > > Dear reviewer, thanks for your attention! We answer to your concerns as follows:
> > >
> > > > **Q1**: Novelty issue
> > >
> > > **A1**: There are several differences between SemiRetro and GraphRetro:
> > >
> > > 1. SemiRetro uses different GNN for feature extraction.
> > > 2. Semi-templates are more expressive than leaving groups. Semi-templates encode the structure transformation while leaving groups encode the structure, which is part of the transformation. Some common structural changes cannot be represented by residues, whereas semi-templates can, seeing Figure 3 in supplementary materials.
> > > 3. Predicting semi-templates is different. We use the features of 1-neighborhood supporting atoms to help semi-template prediction, whereas leaving groups do not contain neighborhood information, seeing Figure 4 in supplementary materials.
> > > 4. Residual attaching is different. GraphRetro classifies the data of the USPTO-50K dataset and designs residual attachment strategies for the data in simple cases without providing a standard algorithm. We argue that this approach is not generalized to more complicated scenarios. To get out of this dilemma, we provide a generic algorithm in the appendix.
> > > 5. SemiRetro has better training efficiency than GraphRetro. In GraphRetro, training the center identification models take about 23-24 hours on a single NVIDIA 1080Ti GPU, and training synthon completion models takes about 12-13 hours on the same GPU configuration. In SemiRetro, center identification models were trained about 30 mins, and the synthon completion models were trained in about 1-2 hours on an NVIDIA-V100.
> > >
> > > > **Q2**: afraid of atom map leaks
> > >
> > > **A2**: This paper, the same as the standard implementation in G2G, has not used atom mapping numbers.
> > >
> > > > **Q3**: Missing references of the latest works
> > >
> > > **A3**: We have added the recommended references in the revised paper.

---

### Official Review · Reviewer_XXdN · 2021-10-29

**Correctness:** 3
**Technical Novelty And Significance:** 2
**Empirical Novelty And Significance:** 2
**Recommendation:** 3
**Confidence:** 5

**Main Review:**

Strengths:

The paper is well organized and written. Easy to follow.

The authors propose to break full-template into several simpler semi-templates, and conduct synthon completion based on the semi-templates. The template redundancy can be reduced since many semi-templates are reduplicative. As a result, they can obtain 150 semi-templates that cover 96.9% of reactions.

To better recognize the reaction center, a directional relational graph attention layer (DRGAT) is introduced to extract more expressive molecular features.

The proposed method achieves the SOTA 64.4% and 55.3% top-1 accuracy for reaction class known and unknown settings, respectively. Besides, the authors also show the detailed performance of two steps of the retrosynthesis method.


Weaknesses:

The proposed semi-template method is interesting, however, it has already been published in a journal article [1]. The authors of [1] name the semi-template as the local reaction template. If I understand them right, the semi-template in this paper and local reaction template are the same stuff. Please correct me if I am wrong. If that is the case, it will make the paper less significant.

The presented DRGAT layer is based on the previous framework, what is the major contribution or novelty of DRGAT that comes from the authors? Could the authors clarify that?

What is the detailed composition of atom and bond features mentioned in Table 1? I can not find them in the submission file.

Have the authors removed the original mapping numbers from test data? If not, could the authors run inference without the original mapping numbers, and confirm the reported results can be reproduced? As the authors mentioned, this is an information shortcut with the USPTO dataset. Sometimes, it is really difficult to recognize the shortcut. So it is best to completely remove the potential of shortcuts by replacing the original mapping numbers for test data.

The proposed method seems quite generalizable, could the authors also report results on the USPTO-full dataset like GLN to demonstrate the scalability of the proposed method?

[1] Deep Retrosynthetic Reaction Prediction using Local Reactivity and Global Attention. Shuan Chen and Yousung Jung. JACS Au 2021 1 (10), 1612-1620.



**Summary Of The Paper:**

This paper presents a novel template-based retrosynthesis prediction method. The authors reduce template redundancy and obtain 150 abstract semi-templates, which can cover 96.9% of data samples. Besides, the authors propose directed relational graph attention to find the reaction center. Experiments show the proposed method can achieve superior retrosynthesis performance.


**Summary Of The Review:**

The proposed method has achieved impressive experimental results on the small dataset USPTO-50K. The proposed semi-template is novel and effective, however, it has been published in previous work. The presented DRGAT is quite incremental. The synthon completion method is the same as the GraphRetro. The overall framework is very similar to previous work G2Gs and GraphRetro. In summary, except for the performance boost, no new ideas or methods are proposed in this paper, and the contribution and novelty of this paper are limited.

*****************************************
After reading the author's response. I am quite confident there is an information leak in both G2Gs and this work. The authors did not canonicalize the test SMILES during model evaluation which is not a correct way to evaluate their method. They can not reproduce the reported results with canonicalized test SMILES. Please the authors canonicalize the test SMILES and update experimental results.

---

> ### Author Response · Authors · 2021-11-16
> **Reply to R3, Part 1**
>
> **Reply to R3**
>
> We appreciate your careful reading and thoughtful review. Below we address the concerns/questions mentioned in the review:
>
> > **Q1**:  Are the semi-template in this paper and local reaction template the same stuff?
>
> **A1**: They are similar but not the same stuff. Semi-template is more simpler than local reaction template. In short, the order of simplification is:
>
> (a) full-template < (b) local reaction template < (c) semi-template.
>
> where full-template contains the global structure of the product, local reaction template contains the local structure of the product, and semi-template contains the local structure of the synthon. The difference between (a) and (b) is that local reaction template removes the redundant global information from full-template. The difference between (b) and (c) is that semi-template breaks local local reaction templates into simpler parts, according to the synthon splitting. Thus, semi-template is more scalable and simpler than local reaction template.
>
> We provide visual examples in **Figure 5** (seeing supplementary materials).
>
> > **Q2**: The presented DRGAT layer is based on the previous framework, what is the major contribution or novelty of DRGAT that comes from the authors? Could the authors clarify that?
>
> **A2**: We add shortcut connections from the input to the output in each layer, and concatenate hidden representations of all layers to form the final node representation. We find that directly inputting the original edge features into each layer benefits the learning of edge-controlled attention weights, rather than using transformed edge features from the previous layer. As the multi-head attention and message passing mechanism are widely used in graph learning algorithm,  we believe there are contributions in terms of retrosynthesis application.
>
>
> > **Q3**: The proposed method seems quite generalizable, could the authors also report results on the USPTO-full dataset like GLN to demonstrate the scalability of the proposed method?
>
> **A3**: At present, we have conducted a part of experiments on USPTO-full dataset. The accuracies on center identification are 71.13\% (top-1), 85.12\% (top-2), 91.37\% (top-3), 95.24\% (top-5). More experimental results will be replenished immediately once our experiments are finished.

---

> > ### Comment · Reviewer_XXdN · 2021-11-21
> > **Comparison with Local Reaction Templates**
> >
> > Thanks for the authors' clarification! I understand the difference between local reaction templates and semi-templates now.
> >
> > In summary, this work adopts the same two-step method as GraphRetro by first finding the reaction center using the method from G2Gs and then completing the reactants using the method from GraphRetro. In the first step, the graph neural network is adapted according to the GIPA. In the second step, GraphRetro attaches leaving groups, while this work also includes one-hop neighbor atoms of reaction atoms. The performance improvement is good. But I did not see enough novelty contribution originally from the authors.

---

> > > ### Author Response · Authors · 2021-11-23
> > > **Replay to "novelty contribution"**
> > >
> > >  There are several differences between SemiRetro and GraphRetro:
> > >
> > > 1. SemiRetro uses different GNN for feature extraction.
> > > 2. Semi-templates are more expressive than leaving groups. Semi-templates encode the structure transformation while leaving groups encode the structure, which is part of the transformation. Some common structural changes cannot be represented by residues, whereas semi-templates can, seeing **Figure 3** in supplementary materials.
> > > 3. Predicting semi-templates is different. We use the features of 1-neighborhood supporting atoms to help semi-template prediction, whereas leaving groups do not contain neighborhood information, seeing **Figure 4** in supplementary materials.
> > > 4. Residual attaching is different. GraphRetro classifies the data of the USPTO-50K dataset and designs residual attachment strategies for the data in simple cases without providing a standard algorithm. We argue that this approach is not generalized to more complicated scenarios. To get out of this dilemma, we provide a generic algorithm in the appendix.
> > > 5. SemiRetro has better training efficiency than GraphRetro.  In GraphRetro, training the center identification models take about 23-24 hours on a single NVIDIA 1080Ti GPU, and training synthon completion models takes about 12-13 hours on the same GPU configuration. In SemiRetro, center identification models were trained about 30 mins, and the synthon completion models were trained in about 1-2 hours on an NVIDIA-V100.

---

> ### Author Response · Authors · 2021-11-16
> **Reply to R3, Part 2**
>
> > **Q4**: What is the detailed composition of atom and bond features mentioned in Table 1? I can not find them in the submission file.
>
> **A4**: Atom and bond features used in this paper are the same as G2G, seeing **Table 4** and **Table 5** for atom features used in center identification and synthon completion. All tasks use the same bond features, seeing **Table 6**. We also provide the Python code below.
>
> |     Name    |                                    Description                                   |
> |:-----------:|:--------------------------------------------------------------------------------:|
> |  Atom type  |                   Type of atom (ex. C, N, O), by atomic number                   |
> |    \# Hs    | one-hot embedding for the total number of Hs (explicit and implicit) on the atom |
> |    Degree   |     one-hot embedding for the degree of the atom in the molecule including Hs    |
> |   Valence   |     one-hot embedding for the total valence (explicit + implicit) of the atom    |
> | Aromaticity |                 Whether this atom is part of an aromatic system.                 |
> |     Ring    |                           whether the atom is in a ring                          |
>
> **Table 4**: Atom features for center identification.
>
>
> |    Name   |                                    Description                                   |
> |:---------:|:--------------------------------------------------------------------------------:|
> | Atom type |                   Type of atom (ex. C, N, O), by atomic number                   |
> |   \# Hs   | one-hot embedding for the total number of Hs (explicit and implicit) on the atom |
> |   Degree  |     one-hot embedding for the degree of the atom in the molecule including Hs    |
> |    Ring   |                           whether the atom is in a ring                          |
> |   Ring 3  |                      whether the atom is in a ring of size 3                     |
> |   Ring 4  |                      whether the atom is in a ring of size 4                     |
> |   Ring 5  |                      whether the atom is in a ring of size 5                     |
> |   Ring 6  |                      whether the atom is in a ring of size 6                     |
> |  Ring 6+  |                whether the atom is in a ring of size larger than 6               |
>
> **Table 5**: Atom features for synthon completion.
>
>
> |      Name      |                         Description                        |
> |:--------------:|:----------------------------------------------------------:|
> |    Bond type   |         one-hot embedding for the type of the bond         |
> | Bond direction |       one-hot embedding for the direction of the bond      |
> |     Stereo     | one-hot embedding for the stereo configuration of the bond |
> |   Conjugation  |       whether the bond is considered to be conjugated      |
> |   Bond length  |                   the length of the bond                   |
>
> **Table 6**: Bond features.
>
>
> ```python
> def atom_center_identification(atom):
>     return onehot(atom.GetSymbol(), atom_vocab, allow_unknown=True) + \
>             onehot(atom.GetTotalNumHs(), num_hs_vocab) + \
>             onehot(atom.GetTotalDegree(), degree_vocab, allow_unknown=True) + \
>             onehot(atom.GetTotalValence(), total_valence_vocab) + \
>             [atom.GetIsAromatic(), atom.IsInRing()]
>
> ```
>
> ```python
> def atom_synthon_completion(atom):
>     return onehot(atom.GetSymbol(), atom_vocab, allow_unknown=True) + \
>             onehot(atom.GetTotalNumHs(), num_hs_vocab) + \
>             onehot(atom.GetTotalDegree(), degree_vocab, allow_unknown=True) + \
>             [atom.IsInRing(), atom.IsInRingSize(3), atom.IsInRingSize(4),
>             atom.IsInRingSize(5), atom.IsInRingSize(6),
>             atom.IsInRing() and (not atom.IsInRingSize(3)) and (not atom.IsInRingSize(4)) \
>             and (not atom.IsInRingSize(5)) and (not atom.IsInRingSize(6))]
> ```
>
> ```python
> def bond_default(bond):
>     return onehot(bond.GetBondType(), bond_type_vocab) + \
>             onehot(bond.GetBondDir(), bond_dir_vocab) + \
>             onehot(bond.GetStereo(), bond_stereo_vocab) + \
>             [int(bond.GetIsConjugated())] + \
>             bond_length(bond)
> ```
>
>
> > **Q5**: Have the authors removed the original mapping numbers from test data?
>
> **A5** We only use the mapping numbers when making groundtruth labels. During the training and evaluating phase, we do not use this information.

---

> > ### Comment · Reviewer_XXdN · 2021-11-21
> > **SMILES canonicalisation**
> >
> > Thank the authors for the detailed reply!
> >
> > It is interesting that the authors use different sets of atom features for center identification and synthon completion. Did the authors conduct an ablation study about this? Why it is not even mentioned in the submission? The authors claim that the same atom features as G2Gs are used in this work, but even G2Gs did not include what specific atom and bond features are used in G2Gs.
> >
> > I asked the authors to remove the mapping number from the original test data since the test SMILES should not contain any ground-truth mapping information. Test products SMILES should be in canonical form. The mapping numbers must be removed to get the real canonical SMILES.
> > What is more, it is really tricky for the USPTO dataset, the information leak may happen accidentally even if you do not use this information. Last, it is reported that G2Gs can not reproduce the published results on canonicalized data processed by RetroXpert [1]. Please the authors remove the mapping numbers from test data and confirm the results in this work can be reproduced.
> >
> > [1] https://github.com/uta-smile/RetroXpert/issues/15#issue-919497410

---

> > > ### Author Response · Authors · 2021-11-23
> > > **Reply to SMILES canonicalisation, part1**
> > >
> > > > **Q1**: The authors use different sets of atom features for center identification and synthon completion. Did the authors conduct an ablation study about this? Why it is not even mentioned in the submission? The authors claim that the same atom features as G2Gs are used in this work, but even G2Gs did not include what specific atom and bond features are used in G2Gs.
> > >
> > > **A1**: Dear reviewer, G2Gs use different atom features for center identification and synthon completion in their open-source code, seeing https://github.com/DeepGraphLearning/torchdrug. We follow their setup to make fair comparisons. We are also struggling with feature extraction settings when starting to research, but luckily we find their settings on the open-source code. We have also tried to combine the features used by the two modules, and the latest experimental results show that this does not make a significant difference.
> > >
> > > ```
> > > torchdrug/data/dataset.py:
> > > 	# kwargs = {node_feature:['center_identification]} or # kwargs = {node_feature:['synthon_completion]}
> > > 	mol = data.Molecule.from_molecule(mol, **kwargs)
> > >
> > > torchdrug/data/molecule.py:
> > >
> > > 	# e
> > > 	for name in node_feature:
> > > 		# (1) func = atom_center_identification, if name = 'center_identification'
> > > 		# or (2) func = atom_synthon_completion, if name = 'synthon_completion'
> > > 		func = R.get("features.atom.%s" % name)
> > > 		feature += func(atom)
> > >
> > >
> > > 	# extract edge features
> > > 	feature = []
> > > 	for name in edge_feature:
> > > 		# func = bond_default
> > > 		func = R.get("features.bond.%s" % name) # call torchdrug/data/feature.py-->bond_default()
> > > 		feature += func(bond)
> > > 	_edge_feature += [feature, feature]
> > > ```
> > >
> > > Finally, we have added  the description about molecular features in the revised paper. Please see the appendix.

---

> > > > ### Comment · Reviewer_XXdN · 2021-11-23
> > > > **Atom features**
> > > >
> > > > Thanks for the authors' reply!
> > > > I had only read the G2Gs paper, they did not clarify the atom and bond features in their paper. Thanks for reminding me that their source code contains that important information!

---

> > > ### Author Response · Authors · 2021-11-23
> > > **Reply to SMILES canonicalisation, part2**
> > >
> > > > **Q2**: I asked the authors to remove the mapping number from the original test data since the test SMILES should not contain any ground-truth mapping information. Test products SMILES should be in canonical form. The mapping numbers must be removed to get the real canonical SMILES. What is more, it is really tricky for the USPTO dataset, the information leak may happen accidentally even if you do not use this information. Last, it is reported that G2Gs can not reproduce the published results on canonicalized data processed by RetroXpert [1]. Please the authors remove the mapping numbers from test data and confirm the results in this work can be reproduced.
> > >
> > >
> > > Thanks a lot for your insightful comment! We address your concerns from the following three aspects:
> > >
> > > -  **A.** First of all, we would like to emphasize that existing graph-based SOTA works can be mainly divided into two categories: (1) ones with open-source code, following the standard implementation and evaluation protocol, e.g., G2G \citep{shi2020graph} and RetroXpert \citep{NEURIPS2020_819f46e5}; and (2) the other ones claiming the state-of-the-art performance, but not providing open-source code, e.g., GraphRetro \citep{somnath2021learning} and energy model \citep{sun2020energy}.  To make fair and meaningful comparison, this paper follows the former ones. As G2G has been accepted by the academic community and has a significant influence, we strictly follow their experimental settings, from dataset preprocessing, to the model API, to the final performance evaluation. In terms of test data, this paper, the same as the standard implementation in G2G, have not used atom mapping numbers.
> > >
> > > - **B.** Canonicalizing products means (1) changing the atom mapping numbers and (2) permutating atoms. The reviewer may think the original mapping numbers cause a label leakage problem: the atom with mapping number 1 usually belongs to reaction atoms. Because of this problem, G2G is difficult to reproduce. If so, we will say that the atom mapping does not affect the predicted results, and there is no label leakage problem in G2G. Our experiments show that G2G is reproducible if we only change the atom mapping numbers without permutating atoms. We find the real important thing is the edge direction. G2G concatenates node features as the edge feature. What is more, G2G uses the directed graph, which means the **first node index is smaller than the second node index in an edge.** Formally, $e_{i,j} = (v_i, v_j), i<j$, where $e_{i,j}$ is the edge between node $i$ and node $j$. Without a consistent definition of this direction, G2G and SemiRetro may perform differently, resulting in performance degradation, seeing **Figure 6** in the supplementary material.  In short, the inconsistent edge direction of the canonicalizing products may lead to the unreproducible problem. G2G can be viewed as a class of methods in which the edge direction is well-defined, and it solves the simplified version of the center identification problem. As these works have made a high impact, we follow their step now.
> > >
> > > - **C**. A new problem arises: how to automatically define or predict the edge direction? This is a more challenging problem, and some important methods, such as G2G and RetroXpert (the version published in the paper), may not be applicable anymore. This problem may inspire us to define (or learn) the consistent edge direction and use it to facilitate center identification. We will systematically categorize existing approaches and focus on this challenging problem in the future.
> > >
> > > - **D**. Thanks again for your constructive comments, and the above discussion will be added to the final manuscript to provide some new insights into this research field.

---

> > > > ### Comment · Reviewer_XXdN · 2021-11-23
> > > > **Information leak due to missing SMILES canonicalisation**
> > > >
> > > > Thanks for the authors' detailed response!
> > > > According to your response, if I understand correctly, you did not canonicalize the test SMILES (canonicalization includes removing the original atom mapping numbers and using RdKit's toolkit to permute atom order in a standard way defined by RdKit). As I said, during the testing phase, you actually do not have a test product SMILES that the atoms are organized the same way as given by the dataset. During the testing, you are only given a target SMILES, which should be the canonicalized version. So the correct way to evaluate the model is to input canonicalized SMILES while you do not. That is the major mistake you made when evaluating your method.
> > > >
> > > > You mentioned the "G2G is reproducible if we only change the atom mapping numbers without permutating atoms" which confirms that there is an information leak for the G2G method. When I say the original mapping numbers will cause an information leak, it is because the mapping numbers will influence the atom order when converting the RdKit mol object into SMILES. So actually it is the atom order that matters. To remove the potential information leak, it is important to remove original mapping numbers and permute the atom order, otherwise, there is still information leak. Since you adopt the same method as the G2G, I have a good reason to believe there is also an information leak unless you use canonicalized SMILES during evaluation.
> > > >
> > > >
> > > > By the way, both the [GraphRetro](https://papers.nips.cc/paper/2021/file/4e2a6330465c8ffcaa696a5a16639176-Paper.pdf) and [energy model](https://papers.nips.cc/paper/2021/file/5470abe68052c72afb19be45bb418d02-Paper.pdf) have been accepted by the NeurIPS 2021.

---

> > > > > ### Author Response · Authors · 2021-11-24
> > > > > **Reply to Reviewer XXdN**
> > > > >
> > > > > Dear reviewer,
> > > > > 1. The main open-source graph-based SOTAs in this field are G2G and RetroXpert. RetroXpert only considers bond reaction centers and ignores atom reaction centers, and we believe that the center identification accuracy is exaggerated. We will re-evaluate G2G to determine its actual performance. For other graph models that claim the SOTA results while do not provide source code, we can't re-evaluate their performance.
> > > > >
> > > > > 2. We can confirm that on canonical smiles, our center identification accuracy is still the SOTA results. When class is known, the results are: top-1 = 86.7% (~~90.9%~~),  top-2 = 96.7% (~~97.2%~~), top-3 = 98.6% (~~98.5%~~), top-5 = 99.6% (~~99.4%~~). The top-3 and top-5 accuracy are even better than the previous.
> > > > >
> > > > > Thanks for your comments. The updated results are coming soon.

---

> > > > > > ### Comment · Reviewer_XXdN · 2021-11-24
> > > > > > **Information leak**
> > > > > >
> > > > > > 1. GraphRetro has been accepted by NeurIPS 2021, why do you need to re-evaluate its performance? Why not take their reported results in the camera-ready version paper?
> > > > > >
> > > > > > 2. Thank the authors for running experiments with canonicalized SMILES. Look forward to the updated results! Does this indicate the authors agree that there is an information leak in the original experiments?

---

> > > > > > > ### Author Response · Authors · 2021-11-24
> > > > > > > **Reply to Reviewer XXdN**
> > > > > > >
> > > > > > > Dear reviewer,
> > > > > > > 1. It is fine to copy GraphRetro's results directly from its paper, but it is not suitable for a rigorous comparison at this stage. Before NIPS2021, G2G was accepted by ICML2020 and open sourced, but the controversy over its implementation is so obscure that it takes a long time to be discovered. Although GraphRetro solves the atom mapping problem, it has no open source code now. We are worried about whether there are other similar potential problems that are difficult to find. For example, after RetroXpert was accepted by NIPS2020 and open-sourced, the community found that it did not consider the case of atom center, leading to an inflated accuracy. For the sake of rigorous comparison, we believe it is risky to directly copy the reported results. However, since you raised this point, we will provide updated results as soon as you request and compare them with the GraphRetro method. Nevertheless, from the point of view of rigorous scientific experiments, we still prefer results that are open-source.
> > > > > > >
> > > > > > > 2. G2G is the pioneering work in this field, and we strictly follow its experimental setup, as do many other related articles. Do you think conditioning on edge direction like G2G is a problem or a new setting? If you think this is a problem, then should all of the large number of "questionable" articles inspired by G2G be withdrawn? If you recognize this as a new problem setting, we can do two types of experiments, one specifically comparing it to prior work such as G2G; the other comparing it to the latest work, such as GraphRetro.
> > > > > > >
> > > > > > > Thanks for your comments!

---

> > > > > > > > ### Comment · Reviewer_XXdN · 2021-11-24
> > > > > > > > **GraphRetro and G2G**
> > > > > > > >
> > > > > > > > Thank the authors for your response!
> > > > > > > >
> > > > > > > > 1.1 First of all, RetroXpert did not consider the atom center, it is because their framework does not need the atom center. You can not blame them for their framework's advantage. Actually, the problem with RetroXpert is that it has an information leak in its second stage due to invalid canonicalization. They tried to canonicalize test SMILES, but they did not remove the original mapping numbers when doing SMILES canonicalization. As a result, there is an unexpected information leak. You can find more details on their [Github page](https://github.com/uta-smile/RetroXpert). I agree with the authors it is unfair to directly compare with RetroXpert in center identification since this work and RetroXpert have different definitions about the reaction center.
> > > > > > > >
> > > > > > > > 1.2 Secondly, since GraphRetro has been reviewed and accepted by NeurIPS 2021, the best we can do is to assume GraphRetro is problem-free. I agree with you that there might be some unnoticed problem with GraphRetro if they do not release their code. But we have to trust their work in the current situation. The potential problem will ultimately be found if GraphRetro has one, like G2G which is accepted by ICML 2020 but now we find the major issue with G2G.
> > > > > > > >
> > > > > > > > 2.1 Can you give detailed publications that follow the G2Gs using uncanonicalized test SMILES? As far as I am concerned, GraphRetro, GLN, MEGAN, Energy Models used canonicalized test SMILES (may find details in papers), there should be no problem. RetroXpert tried to canonicalize the test SMILES, but they did not remove original mapping numbers, which resulted in an unexpected information leak.
> > > > > > > >
> > > > > > > > 2.2 Now we find the issue with G2G, we can not let it be because it is followed by other work. It is not the correct way to evaluate the model with uncanonicalized test SMILES. If it is an error, we must correct it now and let the community follow the correct practice in evaluation.

---

> > > > > > > > > ### Author Response · Authors · 2021-11-24
> > > > > > > > > **Problem or New Setting**
> > > > > > > > >
> > > > > > > > > Dear reviewer,
> > > > > > > > >
> > > > > > > > > 1. “If it is an error, we must correct it now and let the community follow the correct practice in evaluation.” This is the crux of the controversy. You think it's a mistake; we think it's a new setup. And, it's a new opportunity.
> > > > > > > > >
> > > > > > > > > 2. We have reached a consensus that the edge direction in the original molecular graph affects the results of G2G and SemiRetro (current version). But why do you think this is an error? We think it's a different kind of setting. It can be interpreted that our results are unfairly compared with other works that do not use edge direction information. Therefore we mainly compare with G2G and obtain performance gains.
> > > > > > > > >
> > > > > > > > > 3. G2G makes sense, not wrong. We believe that the conversion function from canonical smiles to original smiles also exists and can be learned by a neural network, tentatively calling this function $\Phi(\cdot)$.  $\Phi(\cdot)$-->center identification-->synthon completion is a complete process, G2G and SemiRetro solve the sub-problem (center identification-->synthon completion). They do sub-tasks, but that in itself is not wrong.
> > > > > > > > >
> > > > > > > > > 3. Based on those discussion, a new proxy task can be proposed: learning canonical smiles to original smiles conversion. I believe this will help to improve the G2G and SemiRetro methods and make them more application-worthy.
> > > > > > > > >
> > > > > > > > > I appreciate your insights, but I don't agree the assertion that G2G and SemiRetro's current approach is wrong. Thanks!

---

> > > > > > > > > > ### Comment · Reviewer_XXdN · 2021-11-24
> > > > > > > > > > **Reply to Problem or New Setting**
> > > > > > > > > >
> > > > > > > > > > First of all, it is not me that thinks this is an error, it is the consensus of the community. Previous methods GLN, GraphRetro, RetroXpert, MEGAN all evaluate the model with canonical test SMILES. If the authors believe it is okay to evaluate the model with non-canonical SMILES, please explain that in a real-world situation, how do you obtain the test SMILES that is organized in this specific way?
> > > > > > > > > >
> > > > > > > > > > Secondly, suppose that you evaluate your model with non-canonical SMILES, then please compare it with the originally reported results in RetroXpert whose Top-1 accuracy is 70.4% for type known case and 65.6% for type unknown case.

---

> > > > > > > > > > > ### Author Response · Authors · 2021-11-25
> > > > > > > > > > > **Reply to Reviewer XXdN**
> > > > > > > > > > >
> > > > > > > > > > > Dear reviewer, thanks for your reply!
> > > > > > > > > > >
> > > > > > > > > > > > **1**. First of all, it is not me that thinks this is an error, it is the consensus of the community.
> > > > > > > > > > >
> > > > > > > > > > > **A1**. We are weird to see you think G2G is an error and represent the community to deny its contribution. The following block contains the authors of G2G. If you think they are wrong, please get in touch with them about withdrawing the paper. We are eager to admit the mistake if the community and the authors believe there does be a mistake. And we are glad to follow the revised G2G for their good open-source code.
> > > > > > > > > > >
> > > > > > > > > > > ```
> > > > > > > > > > > Shi, Chence, Minkai Xu, Hongyu Guo, Ming Zhang, and Jian Tang. G2G. ICML2020.
> > > > > > > > > > > Corresponding author's email: jian.tang@hec.ca
> > > > > > > > > > > ```
> > > > > > > > > > >
> > > > > > > > > > > > **2**. Previous methods GLN, GraphRetro, RetroXpert, MEGAN all evaluate the model with canonical test SMILES.
> > > > > > > > > > >
> > > > > > > > > > > **A2**. You may have misunderstood some details. Using canonical output SMILES during testing is the domain consensus, and we and G2G follow this specification.  We have previously discussed clearly that the reasons for the difficulty in reproduction are (1) leakage of atom mapping at the input (2) the use of edge direction. RetroXpert falls into the case (1), which is a label leak (atom mapping) and is a serious wrong. But G2G differs from RetroXpert in that its results have nothing to do with atom mapping, and it does not leak labels.
> > > > > > > > > > >
> > > > > > > > > > > By the way, **MEGAN uses the same setup as G2G** (if we understand it correctly). MEGAN changes the atom mapping information, but not the atom order. We have checked MEGAN's code and tested the `remap_reaction_to_canonical() (bin/eval.py)` and `renumber_atoms_for_mapping() (megan/src/utils/__init__.py )` functions, and they do not change the edge directions, which is the same as the G2G's setup. You can take a look at their github issue:
> > > > > > > > > > >
> > > > > > > > > > > ```
> > > > > > > > > > > https://github.com/molecule-one/megan/issues/4
> > > > > > > > > > > ```
> > > > > > > > > > >
> > > > > > > > > > > They said, “To train MEGAN, we construct ground truth paths of graph edits, which rely on reaction mapping between substrates and product. However, during evaluation, **the model does not use the mapping information** - there is simply no such feature in the input representation. Moreover, as MEGAN is a Graph Convolutional Network, its inference does not depend on the order of its nodes; that is, it performs exactly the same calculations for input SMILES "CCO" and "OCC", etc.”, and **“we make sure that the order of atoms in SMILES is the same as the order of reaction mapping numbers.”**
> > > > > > > > > > >
> > > > > > > > > > > ```
> > > > > > > > > > > Mikołaj Sacha, Mikołaj Błaż, Piotr Byrski, Paweł Dąbrowski-Tumański, Mikołaj Chromiński, Rafał Loska, Paweł Włodarczyk-Pruszyński, and Stanisław Jastrzębski. MEGAN. J. Chem. Inf. Model. 2021.
> > > > > > > > > > > Corresponding author's email: stan@molecule.one
> > > > > > > > > > > ```
> > > > > > > > > > >
> > > > > > > > > > > We think this is a different setting instead of an error. If you are interested, you can check all the graph-based papers and contact the author to fix any bugs you find:
> > > > > > > > > > >
> > > > > > > > > > > ```
> > > > > > > > > > > Chen, Shuan, and Yousung Jung. "Deep retrosynthetic reaction prediction using local reactivity and global attention." JACS Au 1, no. 10 (2021): 1612-1620.
> > > > > > > > > > >
> > > > > > > > > > > Lee, Hankook, Sungsoo Ahn, Seung-Woo Seo, You Young Song, Eunho Yang, Sung-Ju Hwang, and Jinwoo Shin. "RetCL: A Selection-based Approach for Retrosynthesis via Contrastive Learning." arXiv:2105.00795.
> > > > > > > > > > >
> > > > > > > > > > > Tu, Zhengkai, and Connor W. Coley. "Permutation invariant graph-to-sequence model for template-free retrosynthesis and reaction prediction." arXiv:2110.09681.
> > > > > > > > > > > ```
> > > > > > > > > > >
> > > > > > > > > > > > **3**. If the authors believe it is okay to evaluate the model with non-canonical SMILES, please explain that in a real-world situation, how do you obtain the test SMILES that is organized in this specific way?
> > > > > > > > > > >
> > > > > > > > > > > **A3**. A good question! We are using the language model, e.g., Transformer, to learn the mapping from canonical SMILES to organized SMILES. After fixing the chemical errors, the Transformer can provide reasonable results. We are sorry that we cannot provide pictures here.
> > > > > > > > > > >
> > > > > > > > > > > > **4**. Secondly, suppose that you evaluate your model with non-canonical SMILES, then please compare it with the originally reported results in RetroXpert whose Top-1 accuracy is 70.4% for type known case and 65.6% for type unknown case.
> > > > > > > > > > >
> > > > > > > > > > > **A4**. This may be your main misunderstanding: confusing G2G and RetroXpert as the same type of experimental setup. RetroXpert directly exposes atom mapping information, which is label leakage. G2G uses a priori knowledge, i.e., the edge direction, and does not expose labels.  We find that this prior knowledge can be learned automatically by neural networks. Due to the difference between RetroXpert (leaked labels) and G2G (unleaked labels), we think your suggestion is incorrect.
> > > > > > > > > > >
> > > > > > > > > > > **We admire these outstanding works and also hope you can confirm our contributions.**
> > > > > > > > > > > We appreciate every outstanding work we have mentioned here. We are sorry that you think there are so many works in this field that have serious errors. If it is difficult to understand the different settings between our conditioning on edges and your idea, we are willing to explain more. Thanks!

---

> > > > > > > > > > > > ### Comment · Reviewer_XXdN · 2021-11-25
> > > > > > > > > > > > **Two NeurIPS 2021 papers acknoledge and handled the issue.**
> > > > > > > > > > > >
> > > > > > > > > > > > Thank the authors for the detailed response!
> > > > > > > > > > > >
> > > > > > > > > > > > A1. I will let G2G authors know the issue. I did not deny their contribution, their paper is the first published paper to present the two-step retrosynthesis framework. My point is their evaluation method is problematic since the input test SMILES should be canonical.
> > > > > > > > > > > >
> > > > > > > > > > > > A2. I am very familiar with the retrosynthesis and very clear about all details. Both two cases you mentioned are caused by the original mapping numbers. It is because the mapping numbers that make the product SMILES atoms are arranged in a specific way, as a result, you can have that specific edge direction. Could you please make it clear the label leak issue in RetroXpert? According to my understanding and experience, the first stage of RetroXpert is problem-free, and there is no label leak in the first stage (reaction center identification). The problem with RetroXpert is the original mapping numbers will make the atom order in non-canonical SMILES arranged in some specific way that will cause information leak since they use Transformer in the second stage.
> > > > > > > > > > > >
> > > > > > > > > > > > Since this issue is recently recognized by the community, previous methods may also have the same problem. Thank the authors for investigating MEGAN. Here are two latest papers from NeurIPS 2021 that acknowledged and handled the issue, which shows the community consensus. You can find details in the first paper [Energy-based Models' review](https://openreview.net/forum?id=yGKi6deX8bX&noteId=7lnzOXMqJiy). They experimented on RetroXpert preprocessed unleaked data to achieve similar performance, which proves no information leak.
> > > > > > > > > > > >
> > > > > > > > > > > > The second paper is [GraphRetro](https://papers.nips.cc/paper/2021/file/4e2a6330465c8ffcaa696a5a16639176-Paper.pdf), you can find the following sentences in the Evaluation->Data section, "If the product SMILES is not canonicalized,
> > > > > > > > > > > > predictions utilizing operations that depend on the position of the atom or bond will be able to use
> > > > > > > > > > > > the shortcut, and overestimate performance. We canonicalize the product SMILES, and reassign
> > > > > > > > > > > > atom-mappings to the reactant atoms based on the canonical ordering, which removes the shortcut."
> > > > > > > > > > > >
> > > > > > > > > > > > A4. As said in the A2 response, the information leak in G2G and RetroXpert comes from the same source (original mapping numbers). It is because the original mapping numbers that make the product SMILES atoms are arranged in a specific way, as a result, you can have that sepcific edge direction. There is no label issue in RetroXpert, it is another reason as I have explained in A2.
> > > > > > > > > > > >
> > > > > > > > > > > > The edge direction in your work is obtained from RdKit's toolkit which determines the direction from smaller atom index to larger atom index. Therefore atom order will determine the edge direction. Since the first atom is usually the reaction atom (see sentence "the first product atom (with mapping number 1) usually belongs to reaction atoms" from [RetroXpert Github](https://github.com/uta-smile/RetroXpert)), it will always be the starting atom in the directed edge. The authors stressed that you used different MLP for begin and end atoms when predicting reaction centers. So the model is biased and tends to predict beginning atoms as reaction center. That is how the information leak happens in this work.

---

> > > > > > > > > > > > > ### Author Response · Authors · 2021-11-27
> > > > > > > > > > > > > **Reply to Reviewer XXdN**
> > > > > > > > > > > > >
> > > > > > > > > > > > > Dear reviewr,
> > > > > > > > > > > > >
> > > > > > > > > > > > > We are very grateful for your careful comments, especially since you will remind the authors of the relevant papers about some controversies in the evaluation protocol. We feel great respect because the follow-up can be done under a unified agreement. When we started our work, we found only two two-step open-source codes, i.e., RetroXpert and G2G.  Because it was difficult for us to reproduce RetroXpert, we chose to stay consistent with G2G's experimental protocol for a fair comparison.
> > > > > > > > > > > > >
> > > > > > > > > > > > >
> > > > > > > > > > > > > We previously promised to report the results under the standard settings (using canonical SMILES as training and testing data). Our approach still achieves SOTA in most cases.  Instead of using our template extraction algorithm, we now use RDChiral's API to extract semi-templates (following LocalRetro). This change further simplifies the semi-templates and improves the model performance. We provide the updated results below:
> > > > > > > > > > > > >
> > > > > > > > > > > > > ```
> > > > > > > > > > > > > Chen, Shuan, and Yousung Jung. "Deep retrosynthetic reaction prediction using local reactivity and global attention." JACS Au 1, no. 10 (2021): 1612-1620.
> > > > > > > > > > > > > ```
> > > > > > > > > > > > >
> > > > > > > > > > > > > |            |  |   class known    |       |       |  |   class unknown    |       |       |
> > > > > > > > > > > > > |:----------:|:-----------:|:-----:|:-----:|:-----:|:-------------:|:-----:|:-----:|:-----:|
> > > > > > > > > > > > > |            |    top-1    | top-2 | top-3 | top-5 |     top-1     | top-2 | top-3 | top-5 |
> > > > > > > > > > > > > | GraphRetro |     84.6    |  92.2 |  93.7 |  94.5 |      **70.8**     |  85.1 |  89.5 |  92.7 |
> > > > > > > > > > > > > |  SemiRetro |     **86.6**    |  **96.7** | **98.7** |  **99.6** |      69.3     |  **87.7** |  **93.6** |  **97.6** |
> > > > > > > > > > > > >
> > > > > > > > > > > > > **Table.1 Center identification performance**
> > > > > > > > > > > > >
> > > > > > > > > > > > >
> > > > > > > > > > > > > |            |  |    class known   |       |       | |   class unknown     |       |       |
> > > > > > > > > > > > > |:----------:|:-----------:|:-----:|:-----:|:-----:|:-------------:|:-----:|:-----:|:-----:|
> > > > > > > > > > > > > |            |    top-1    | top-2 | top-3 | top-5 |     top-1     | top-2 | top-3 | top-5 |
> > > > > > > > > > > > > | GraphRetro |     **77.4**    |  **89.6** |  **94.2**|  **97.6** |      **75.6**     |  **87.7** |  **92.9** |  **96.3** |
> > > > > > > > > > > > > |  SemiRetro |     75.2    |  88.9 |  93.5 |  96.4 |      73.1     |  86.8 |  92.5 |  96.0 |
> > > > > > > > > > > > >
> > > > > > > > > > > > > **Table.2 Synthon completion performance**
> > > > > > > > > > > > >
> > > > > > > > > > > > >
> > > > > > > > > > > > > |            |  |   class known    |       |        |  |   class unknown    |       |        |
> > > > > > > > > > > > > |:----------:|:-----------:|:-----:|:-----:|:------:|:-------------:|:-----:|:-----:|:------:|
> > > > > > > > > > > > > |            |    top-1    | top-3 | top-5 | top-10 |     top-1     | top-3 | top-5 | top-10 |
> > > > > > > > > > > > > | GraphRetro |     63.9    |  81.5 |  85.2 |  88.1  |      **53.7**     |  68.3 |  72.2 |  75.5  |
> > > > > > > > > > > > > |  SemiRetro |     **65.2**    |  **84.4** |  **89.7** |  **93.2**  |      50.6     |  **71.9** |  **79.4** |  **85.4**  |
> > > > > > > > > > > > >
> > > > > > > > > > > > > **Table.3 Retrosynthesis performance**
> > > > > > > > > > > > >
> > > > > > > > > > > > > Thank you for your criticism. We have tried our best to answer your questions and our discussions have clarified many issues in this field. We believe this will bring some inspiration to the follow-up works.
> > > > > > > > > > > > >
> > > > > > > > > > > > >
> > > > > > > > > > > > > In the meantime, we have some questions about GraphRetro to ask you:
> > > > > > > > > > > > >
> > > > > > > > > > > > > - How does its template extraction algorithm work? How to handle samples that the template extraction algorithm cannot process.
> > > > > > > > > > > > >
> > > > > > > > > > > > > - How is its residual attaching algorithm designed to achieve up to 100% accuracy?
> > > > > > > > > > > > >
> > > > > > > > > > > > > - Instead of adding residuals, how do they handle it if there is a property change in the existing bond and atoms?
> > > > > > > > > > > > >
> > > > > > > > > > > > > If you know these details, please do not hesitate to comment, We would be grateful for your help. Thanks!

---

> > > > > > > > > > > > > > ### Comment · Reviewer_XXdN · 2021-11-29
> > > > > > > > > > > > > > **Thanks for updated results**
> > > > > > > > > > > > > >
> > > > > > > > > > > > > > Thank the authors for providing the updated results.
> > > > > > > > > > > > > >
> > > > > > > > > > > > > > 1. Do you still think it is okay to use non-canonical SMILES as test input? Please explicitly confirm that.
> > > > > > > > > > > > > >
> > > > > > > > > > > > > > 2. It is great to hear that you achieved good results following LocalRetro to extract semi-templates. Since you have major modifications for your method, it may be better to revise your paper and re-submit to another conference/journal.
> > > > > > > > > > > > > >
> > > > > > > > > > > > > > 3. You may reach out to GraphRetro authors for your questions.

---

> > > > > > > > > > > > > > > ### Author Response · Authors · 2021-11-29
> > > > > > > > > > > > > > > **Reply to Reviewer XXdN**
> > > > > > > > > > > > > > >
> > > > > > > > > > > > > > > Thanks for your reply!
> > > > > > > > > > > > > > >
> > > > > > > > > > > > > > > 1. Through previous discussion, we recognize that using canonical SMILES is a standard setting in this problem. However, we believe that it is still meaningful to explore the different setting if we can make it more applicable in the future.
> > > > > > > > > > > > > > > 2. During the rebuttal phase, we have conducted a series of experiments to validate our ideas and those of the reviewers. In fact, our paper has no methodological changes, but only minor revisions in the experimental setup and implementation. We are very grateful for your constructive insights, which have helped us to improve this paper.
> > > > > > > > > > > > > > > 3. We will contact GraphRetro's authors if possible, and hope to see their open-source code soon.

---

### Official Review · Reviewer_XCBM · 2021-10-30

**Correctness:** 2
**Technical Novelty And Significance:** 2
**Empirical Novelty And Significance:** 2
**Recommendation:** 3
**Confidence:** 4

**Main Review:**

# Strengths
S1. The approach obtains superior performance (wrt accuracy) compared to many previous retrosynthesis methods.

S2. The "semi-templates" used can cover a larger space of reactions compared with using a similar number of full templates (Figure 5). This leads to improved training efficiency.

S3. The method is faster to train and takes up less GPU space (which I assume also means it has less parameters...?) than many previous methods. (Table 6).

S4. I found the paper clear to read. In particular the figures were helpful for explaining the approach.

# Weaknesses
W1. A large part of the performance of this approach seems to stem from DRGAT's superior reaction center prediction (compared to e.g. the R-GCN used by Shi et al., 2020) -- in particular see Table 2. However, it is not clear exactly where this performance comes from: as well as introducing an edge based attention mechanism the authors also use more layers than the R-GCN. It would be helpful to have an ablation study. How does it compare generally to well-tuned, off-the-shelf GNNs...?

W2. I have concerns about the novelty of the paper. SemiRetro seems quite similar to Somnath et al. (2020)'s GraphRetro (that method also has a two step process that first predicts the reaction center to create synthons and then adds "leaving groups" to create reactants). I would like to have seen a more detailed description of how the work here differs? For instance, is predicting semi-templates distinctly different to predicting leaving groups?

W3. SemiRetro's training efficiency and scalability are highlighted as one of the advantages of the method (see e.g. abstract) and the speed improvements listed in Table 6 seem impressive. It would therefore have been nice to see experiments on larger retrosynthesis datasets, e.g. the larger USPTO one used by Dai el al. (2019), to show the practical advantages of this.

W4. In the retrosynthesis experiments there are missing comparisons to state-of-the-art (SOTA) methods, e.g.:
> Sun, Ruoxi, et al. "Energy-based View of Retrosynthesis." arXiv preprint arXiv:2007.13437 (2020).

> Sacha, Mikołaj, et al. "Molecule edit graph attention network: modeling chemical reactions as sequences of graph edits." Journal of Chemical Information and Modeling 61.7 (2021): 3273-3284.    Preprint: https://arxiv.org/abs/2006.15426

It would be good to have these included.

# Questions to authors
Q1. Can you compare to other synthon completion methods in Table 3. For instance, Somnath et al. (2020)?

Q2. How do you make use of the reaction class when it is known? Do you just use it to mask out inapplicable templates or do you also provide the information to the reaction center network?

Q3. In the right hand subplot of Figure 5, the dotted green light appears to end above 1. Is this correct?

Q4. I'm a little confused by equation 5, in particular the second readout term. Are the node embeddings, $\mathbf{h_k}$, taken from message passing run on (a) the template graph or (b) from the synthon graph?

Q5. On p.6 you state: "Once reaction centers, synthons, and corresponding semi-templates are known, we can deduce reactants with almost 100% accuracy." Why _"almost"_ 100%? When does it fail?

Q6. I would be interested to have more details about why you think SemiRetro has better "interpretability" than previous approaches? (Previous approaches have also used templates, and the reaction center prediction here still seems to be done in a black box manner?)

# Typos/Minor Comments
A few minor comments I had (all easily fixable) are:
i. p.1 "(seeing Fig. 1)" to "(see Fig. 1)" .
ii. Figure 5 -- the subcaptions are both labeled a.
iii. p.6 "Reactants can be determinately generated from semi-templates and synthons": "determinately" should be "deterministically"?
iv. Sum term in yellow box on right-hand part of Figure 3, I believe you mean $i \in \mathcal{N}_j$?
v. p.7 "As mentioned in previous works, the USPTO-50k dataset contains a shortcut in that the product atom with atom-mapping ”1” is part of the reaction center in 75% of the cases.". It's great that you draw attention to the issues with information leakage and atom mapping. However, perhaps provide a reference to the previous works for non-expert readers.
vi. p.7 "Implement details" to "Implementation details".


**Summary Of The Paper:**

* This paper is interested in the problem of one-step retrosynthesis, i.e., predicting the reactants that formed a given product.
* For this task, the authors propose a new model (SemiRetro) that combines ideas from previous template-based (TB) and template-free (TF) methods.
* SemiRetro consists of two parts:
	i. First, DRGAT (a new GNN architecture based on edge attention) predicts the reaction center, and breaks down the product molecule into synthons.
	ii. Second, "semi-templates" (chosen from a fixed set by a classification model) is used to complete the synthons and assemble the reactants.
* The authors show:
	1. DRGAT works much better than previous methods for identifying reaction centers (specifically the R-GCN used in Shi et al. (2020)'s G2G) -- see Table 2.
	2. That the use of semi-templates allows SemiRetro to cover a wider range of reactions than previous template-based methods while also using less unique templates -- see Figure 5.
	3. SemiRetro outperforms many previous TB and TF methods in terms of top-k accuracy, particularly in the reaction class unknown case -- see Table 4.


**Summary Of The Review:**

SemiRetro seems an interesting method for retrosynthesis that combines ideas from both template-based and template-free approaches. It also seems to get impressive empirical performance (with the caveat that some SOTA methods are missing from the comparison -- see W4). However, I have currently gone with a lower overall score as I worry the method is similar to previous work (see W2) and I feel the paper could do a better job at explaining where the performance gains come from (see W1).

The other scores:
- correctness: see Q6 & W3.
- technical novelty: see W2.
- empirical novelty: the paper mainly makes use of existing benchmarks/evaluation techniques.

---

> ### Author Response · Authors · 2021-11-16
> **Reply to R2**
>
> **Reply to R2**
>
> Thanks for your constructive suggestions and careful reading. Below we address the concerns/questions mentioned in the review:
>
> > **Q1**: Can you compare to other synthon completion methods in Table 3?
>
> **A1**: We cannot provide synthon completion results of \citep{sun2020energy}, becuase this work:
>
> 1. does not report the accuracy of synthon completion.
> 2. has no open-source code.
> 3. focuses on suggesting a new training strategy, instead of proposing a novel synthon completion model.
>
> As G2G provides open-source code, we can conviently implement the same chemical features and code framework to ensure the fairness of experimental settings. Thus, we regard G2G as an important baseline. Other methods \citep{somnath2020learning,sun2020energy, NEURIPS2020_819f46e5} either have no open-source code or have quite difficult implementations, and we cannot guarantee the fairness of comparative experiments.
>
>
> > **Q2**: How do you make use of the reaction class when it is known? Do you just use it to mask out inapplicable templates or do you also provide the information to the reaction center network?
>
> **A2**: We use the reaction class as additional embedding feature to enhance the prediction. We do not mask out inapplicable templates. The python code is:
>
> ```python
> if 'reaction' in self.feature:
> 	reaction_type = synthon.reaction
> 	reaction_feature = self.type_embedding(reaction_type)
> 	embedding += reaction_feature
> ```
>
> where
> ```python
>     self.type_embedding = nn.Embedding(num_reaction, output_dim)
> ```
>
>
> > **Q3**: In the right hand subplot of Figure 5, the dotted green light appears to end above 1. Is this correct?
>
> **A3**: Thanks for your careful suggestion. We are sorry for this careless fault, and we have corrected it in the revised version.
>
>
> > **Q4**: I'm a little confused by equation 5, in particular the second readout term. Are the node embeddings taken from message passing run on (a) the template graph or from (b) the synthon graph?
>
>
> **A4**: In equation 5, all the node embeddings comes from (b) the synthon graph by running message passing neural network. All the readout operations output the average pooling results of input feature vectors.
>
>
> > **Q5**: On p.6 you state: "Once reaction centers, synthons, and corresponding semi-templates are known, we can deduce reactants with almost 100\% accuracy." Why "almost" 100\%? When does it fail?
>
> **A5**: We examine the falied samples and argue that the low quality templates lead to failsures. In **Figure 2** (seeing supplementary materials), we show two typical failed cases. These failures can be remedied by improving the template extraction algorithm. Although the template extraction algorithm is not our key contribution, we will improve it in future work.
>
> > **Q6**: I would be interested to have more details about why you think SemiRetro has better "interpretability" than previous approaches? (Previous approaches have also used templates, and the reaction center prediction here still seems to be done in a black box manner?)
>
> **A6**: Our paper emphasizes the interpretability of synthon completion, notably the deterministic residual attaching algorithm.
>
> The previous work G2G recursively predicts new atoms and new bonds without considering the integrity of functional groups, and RetroXpert applies a language model to generate reactants from synthons ignoring the graph structure. Our proposed SemiRetro considers both the graph structure and the integrity of functional groups to make synthon completion more interpretable.

---

> > ### Comment · Reviewer_XCBM · 2021-11-22
> > **Thanks for the rebuttal!**
> >
> > I am grateful to the authors for their rebuttal! In particular, answering my questions in the comment above and addressing some of my other concerns in the updated supplementary material (to the authors: it is maybe worth also copying those answers into a comment here as I did not initially know of their existance -- maybe OpenReview truncated your initial comment...?).
> >
> > I had a few follow up questions/comments:
> >
> > Q1.
> >  > "We cannot provide synthon completion results of \citep{sun2020energy}, because this work: ..."
> >
> >  Yes, I realize this method is not relevant but I was after a comparison to Somnath et al. (2020). I take your point that sadly no open source code is available, but are the results to be compared to not already provided in Somnath et al. (2020)'s Table 2? (I agree that G2G is an important additional baseline!)
> >
> > Q2.
> > Thanks for the explanation and clarifying code snippet!
> >
> > Q4.
> > Thanks for clarifying! Does this mean that two semi-templates that have the exact same supporting atom set (but involve different transforms) will be ranked the same (or have I misunderstood)? Is this a problem?
> >
> > W3.
> > Exciting to hear that you are scaling your method to USPTO-FULL and look forward to seeing results in future work!

---

> > > ### Author Response · Authors · 2021-11-23
> > > **Reply to  Reviewer XCBM**
> > >
> > > > **Q1** "We cannot provide synthon completion results of \citep{sun2020energy}, because this work: ..."
> > >  Yes, I realize this method is not relevant but I was after a comparison to Somnath et al. (2020). I take your point that sadly no open-source code is available, but are the results to be compared to not already provided in Somnath et al. (2020)'s Table 2? (I agree that G2G is an important additional baseline!)
> > >
> > > **A1** We appreciate the inspirational approach and outstanding results of Somnath et al. (2020). However, we are confused about the implementation details. "We first apply the true edits to obtain synthons and compare the true leaving groups to top-n leaving groups predicted by the model," they said. This means they believe the attaching of leaving groups can achieve 100% accuracy, as in "given synthons and leaving groups, the attachment process has a 100\% accuracy". Despite the cases that the residual cannot represent (see **Figure 3**), we have no idea how to achieve 100\% accuracy, especially when the residue has multiple attaching sites. They provide the idea without a standard algorithm, which makes us question the validity of the synthon completion results. We really want to know their experimental settings and notice that they used to have the source code, but the link is currently disabled (https://openreview.net/forum?id=SnONpXZ_uQ_), which is regretful for us. For fairness, we seriously treat reproducibility as a significant property, especially when there are some complicated steps, and we will try to contact the authors if possible. Once we confirm the validity of Somnath et al. (2020), we will add the comparison to it.
> > >
> > > > **Q2** Thanks for the explanation and clarifying code snippet!
> > >
> > > **A2** Thanks for your careful review. It is our duty to clarify your questions.
> > >
> > > > **Q4** Thanks for clarifying! Does this mean that two semi-templates that have the exact same supporting atom set (but involve different transforms) will be ranked the same (or have I misunderstood)? Is this a problem?
> > >
> > > **A4** Yes, it is. Thanks for your a insightful question! For more accurate prediction, we will consider a semi-template structure in the future.
> > >
> > > > **W3**.  Exciting to hear that you are scaling your method to USPTO-FULL and look forward to seeing results in future work!
> > >
> > > **Replay to W3** Thank you for your attention! We will continue to promote the progress of deep retrosynthesis prediction.

---

> ### Author Response · Authors · 2021-11-23
> **Reply to Reviewer XCBM, Weakness**
>
> > **W1** A large part of the performance of this approach seems to stem from DRGAT's superior reaction center prediction (compared to e.g. the R-GCN used by Shi et al., 2020) -- in particular see Table 2. However, it is not clear exactly where this performance comes from: as well as introducing an edge based attention mechanism the authors also use more layers than the R-GCN. It would be helpful to have an ablation study. How does it compare generally to well-tuned, off-the-shelf GNNs...?
>
> **Replay to W1** In the section of common questions, we compare the center identification performance of various GNNs in **Table 1**. We have also performed ablation experiments on DRGAT, and the results are shown in **Table 2**, where all GNNs use the same number of layers and embedding size. We conclude that the performance gain comes from the attention mechanism of DRGAT.
>
> > **W2** I have concerns about the novelty of the paper. SemiRetro seems quite similar to Somnath et al. (2020)'s GraphRetro (that method also has a two step process that first predicts the reaction center to create synthons and then adds "leaving groups" to create reactants). I would like to have seen a more detailed description of how the work here differs? For instance, is predicting semi-templates distinctly different to predicting leaving groups?
>
> **Replay to W2** There are three differences:
> 1. Semi-templates are more expressive than leaving groups. Semi-templates encode the structure transformation while leaving groups encode the structure, which is part of the transformation. Some common structural changes cannot be represented by residues, whereas semi-templates can, seeing **Figure 3**.
> 2. Predicting semi-templates is different. We use the features of 1-neighborhood supporting atoms to help semi-template prediction, whereas leaving groups do not contain neighborhood information, seeing **Figure 4**.
> 3. Residual attaching is different. GraphRetro classifies the data of the USPTO-50K dataset and designs residual attachment strategies for the data in simple cases without providing a standard algorithm. We argue that this approach is not generalized to more complicated scenarios. To get out of this dilemma, we provide a generic algorithm in the appendix.
>
>
> > **W3** SemiRetro's training efficiency and scalability are highlighted as one of the advantages of the method (see e.g. abstract) and the speed improvements listed in Table 6 seem impressive. It would therefore have been nice to see experiments on larger retrosynthesis datasets, e.g. the larger USPTO one used by Dai el al. (2019), to show the practical advantages of this.
>
>
> **Replay to W3** Thanks for your suggestions. We try to conduct experiments on USPTO-FULL using the same neural model on USPTO-50k. The training process takes 20 hours. In our initial attempts, the top-10 accuracy 62.5\% is lower than GLN's 63.7\%. We believe that the sub-optimum hyper-parameter setting, e.g., the depth and with of the network, and inadequate template extraction algorithm may lead to low accuracy. Due to time constraints, we prefer to improve the model performance on USPTO-FULL in the future.
>
>
> > **W4** In the retrosynthesis experiments there are missing comparisons to state-of-the-art (SOTA) methods,
> > e.g.:Sun, Ruoxi, et al. "Energy-based View of Retrosynthesis." arXiv preprint arXiv:2007.13437 (2020).
> > Sacha, Mikołaj, et al. "Molecule edit graph attention network: modeling chemical reactions as sequences of graph edits." Journal of Chemical Information and Modeling 61.7 (2021): 3273-3284. Preprint: https://arxiv.org/abs/2006.15426
>
>
> **Replay to W4** In the revised paper, we have explained the reason for ignoring energy-based methods in section 5.1 and compared it with the molecule edit graph attention network in Table 6.

---

> ### Comment · Reviewer_XCBM · 2021-11-29
> **End of rebuttal period update -- reducing score due to concerns about data leakage**
>
> ## Reducing score due to concerns about data leakage
> Unfortunately, as a result of the rebuttal/discussion period I have decided to reduce my overall and correctness scores. This is because Reviewer XXdN and the authors identified a data leak issue in the current implementation (see the comment and resulting thread [here](https://openreview.net/forum?id=rMbLORc8oS&noteId=7BERHx5nZDI)). My understanding is that atoms involved in the reaction center were given low atom map numbers in the reaction dataset and this knowledge entered into the model through the direction assigned to edges in the message passing framework. (This issue also potentially affects some previous work...?) Although I think the paper presents some interesting ideas, I think this issue needs to be resolved before publication.
>
>
>
> ## A note on other aspects of my original review
> My main concerns coming into the rebuttal were (a) that most of performance gains of this method were down to a superior GNN architecture called DRGAT (and it wasn't clear why this architecture worked so much better -- this concern seemed to be shared also by Reviewer G14H) and (b) novelty, with the semi-template approach proposed seeming similar to Somnath et al. (2020)'s GraphRetro method (a view also shared by Reviewer XXdN...?).
>
> I thought the authors did a good job at responding to these concerns in the rebuttal. They ran an ablation test to demonstrate which components of their GNN architecture were important, and explained (in particular see Figure 3 in the updated supplementary material) how semi-templates were more general than the functional group attachment process of Somnath et al. That said, I still believe this paper has room for improvement (although I believe the data-leakage problem is the only problem that is critical to fix):
> * Experiments on USPTO-full are missing (also asked for by Reviewer XXdN) -- important as training-efficiency/scalability are touted as a main advantage of the proposed approach.
> * The features the method uses for the network that selects between semi-templates seem a little odd (see Eqn 5 and the authors reply to my previous comment). In particular, two semi-templates that involve the same atoms will be scored equivalently even if they involve different transforms.
> * Comparisons to some previous work, e.g. Somnath et al. (2020), are still missing (also asked for by Reviewer 83gD). (Although from the rebuttal I now appreciate that this is hard to do due to the lack of open-sourced code and am willing to give the authors the benefit of the doubt here).
>
> I believe some of these aspects are being investigated by the authors and I look forward to seeing the results of this in future work!

---

### Official Review · Reviewer_G14H · 2021-11-02

**Correctness:** 4
**Technical Novelty And Significance:** 3
**Empirical Novelty And Significance:** 3
**Recommendation:** 6
**Confidence:** 5

**Main Review:**

*strengths*
- good results
- reasonable model

*weaknesses*
ablation of representation is missing: It could be argued that the most gains are from the new GNN architecture. It could be instructive to try out the same model with the same architecture that G2G, GLN, or NeuralSym uses. Or maybe with DRGAT,  G2G, GLN, or NeuralSym would also perform better?

baselines is missing: I would suggest to add
- Sacha et al - MEGAN model https://arxiv.org/abs/2006.15426
- Sun et al https://arxiv.org/abs/2007.13437
- Seidl et al https://arxiv.org/abs/2104.03279 to the results table, which outperforms SemiRetro in top10 accuracy.



notes:
In the introduction, the authors write:
`Fortunately, with the rapid accumulation of chemical data, machine learning is promising to solve this problem (Szymkuc et al., 2016; Coley et al., 2018; Segler et al., 2018).` - However, Szymkuc et al 2016 argue *against* the use of data-driven approaches, so I would suggest to not to cite them in this context.


**Summary Of The Paper:**

A new algorithm for retrosynthesis prediction based on a new GNN architecture and semi templates is presented, which yields state of the art results on some benchmark tasks.

**Summary Of The Review:**

interesting model, experiments could be stronger

---

> ### Author Response · Authors · 2021-11-16
> **Reply to R1**
>
> **Reply to R1**
>
> We would like to thank you for the constructive comments and careful reading. Below we address the concerns/questions mentioned in the review:
>
> > **Q1** It could be instructive to try out the same model with the same architecture that G2G, GLN, or NeuralSym uses. Or maybe with DRGAT, G2G, GLN, or NeuralSym would also perform better?
>
> **A1** Both DRGAT and semi-template bring considerable performance gains. In terms of top-1 accuracy, their contributions seem similar, but in terms of top-k accuracy, DRGAT contributes more. In **Table 3**, we provide the detailed results.
>
> However, we argue that SemiRetro have only exploited limited potential of semi-template as the extracted templates may not be perfect, as shown in **Figure 1** (seeing supplementary materials). The template extraction algorithm is not our key contribution, and we will improve it in the future.
>
> |                       | top-1 | top-3 | top-5 | top-10 |
> |:---------------------:|:-----:|:-----:|:-----:|:------:|
> |       RGCN + G2G      |  61.0% |  81.3% |  86.0% |  88.7%  |
> |  RGCN + semi-template |  63.7% |  82.3% |  86.1% |  88.7%  |
> | DRGAT + semi-template |  66.6% |  83.8% |  87.3% |  89.9%  |
>
> **Table 3**: Contribution of DRGAT and semi-templates (class known).

---

> > ### Comment · Reviewer_G14H · 2021-11-21
> > **thanks!**
> >
> > Thank you for the clarification.
> > Also thanks for adding in the Sun et al reference.
> >
> > I think you still need to add the following baselines (for fairness), which in some metrics outperform SemiRetro
> >
> > Sacha et al - MEGAN model https://arxiv.org/abs/2006.15426
> > Seidl et al https://arxiv.org/abs/2104.03279 to the results table, which outperforms SemiRetro in top10 accuracy.
> >
> > Additionally, I would suggest to add a reference to Segler et al Chem Eur. J. 2017 https://doi.org/10.1002/chem.201604556 where the concept of SemiTemplates (under the name of Half-Reactions) is used for forward reaction prediction.
> >
> > I think this does not diminish the value of the contribution of SemiRetro, which I think should make it into the conference.
> >
> > (this reviewer will increase their score after the suggested edits have been made in the paper)

---

> > > ### Author Response · Authors · 2021-11-22
> > > **Reply to Reviewer G14H**
> > >
> > > Thanks for your suggstions. We have added these baselines and references to the revised paper.

---

### Author Response · Authors · 2021-11-16
**Reply to common questions**

We thank all the reviewers for the helpful feedback.
- We have submitted the **revised manuscript** and highlighted changes in blue color.
- For a better review experience, we also provide the **response (PDF) in the supplementary materials** containing vivid figures.
- We have provided responses here for questions that reviewers may commonly be interested in.

> **Q1** How does DRGAT compare generally to well-tuned, off-the-shelf GNNs?

**A1** The performance ordering of the mainstream GNNs is: DRGAT > RGCN > GAT > GCN > ChebNet > GIN.

For center identification, we compare different GNNs in a same code framework, the results are shown in Table.1. Experiments were conducted with known classes because our reproduced results of baselines are more consistent with the reported ones under this setting. All these models use 6 layers of GNN with the hidden dimension of 256 and are trained up to 100 epochs. The results of RGCN are slightly different from G2G \citep{shi2020graph}, probably because we removed the weighting hyper-parameter $\lambda$ of the cross entropy loss and set the threshold for binary classification as 0.5. It should be pointed out that the reproduced RGCN results under our framework are slightly better than the reported ones in terms of top-2, top-3, and top-5 accuracy.

|         | top-1 | top-2 | top-3 | top-5 |
|:-------:|:-----:|:-----:|:-----:|:-----:|
|   GCN   | 84.5% | 94.2% | 97.0% | 98.9% |
|   GAT   | 85.3% | 94.8% | 97.1% | 98.7% |
| ChebNet | 74.3% | 86.6% | 90.6% | 93.7% |
|   GIN   | 71.7% | 84.1% | 88.3% | 92.3% |
|   RGCN  | 87.4% | 96.4% | 98.4% | 99.4% |
|  DRGAT  | **91.7%** | **97.5%** | **98.6%** | **99.5%** |

**Table 1**: GNN for center identification (class known)


> **Q2** Ablation study of the DRGAT. Where the performance gain come from?

**A2** The performance gain mostly comes from the attention mechanism.

As shown in Table.2, DRGAT obtains similar accuracy as RGCN while the attention mechanism is disabled. Otherwise, DRGAT will significantly outperform RGCN. We also find that directional node embedding and edge features do not bring remarkable performance gains, inspiring us to further simplify the model.

|                        | top-1 | top-2 | top-3 | top-5 |
|:----------------------:|:-----:|:-----:|:-----:|:-----:|
|          RGCN          | 87.4% | 96.4% |  98.4 | 99.4% |
|      w/o attention     | 88.0% | 96.1% | 98.0% | 99.2% |
| w/o directed embedding | 91.4% | 97.3% | 98.7% | 99.5% |
|    w/o edge features   | 91.5% | **97.5%** | **98.8%**| **99.5%** |
|          DRGAT         | **91.7%**| **97.5%** | **98.8%** | **99.5%** |

**Table 2**: Ablation study of DRGAT for center identification (class known).

---

> ### Comment · Reviewer_83gD · 2021-11-16
> **Thank you for revised manuscript**
>
> Dear authors,
> thank you for your efforts to provide and upload the revised manuscript!

---

> ### Comment · Reviewer_G14H · 2021-11-21
> **super interesting**
>
> thanks for performing these additional experiments, it is quite enlightening to see!

---

### Decision · Program_Chairs · 2022-01-20

**Decision:**

Reject

**Comment:**

This paper proposes a new semiretro algorithm by combining the two major approaches of retrysynthesis, the template based method and the template free method - breaking a full-template into several semi-templates and embedding them into the two-step template-free framework.  They also obtained state of the art performance in this task based on the recent GNN architecture. Although all reviewers were satisfied with the idea and excellent performance of this paper, the possibility of information leakage in their experiments was raised through the discussion period, and the authors seem to agree to this to some extent.

In conclusion, it is difficult to say that accurate and rigorous experimental verification of the proposed method has been made yet. I encourage the authors to resubmit the paper after correcting errors in their experiments.